# Targeting Inflammation by Flavonoids: Novel Therapeutic Strategy for Metabolic Disorders

**DOI:** 10.3390/ijms20194957

**Published:** 2019-10-08

**Authors:** Mohammad Hosein Farzaei, Amit Kumar Singh, Ramesh Kumar, Courtney R. Croley, Abhay K. Pandey, Ericsson Coy-Barrera, Jayanta Kumar Patra, Gitishree Das, Rout George Kerry, Giuseppe Annunziata, Gian Carlo Tenore, Haroon Khan, Matteo Micucci, Roberta Budriesi, Saeideh Momtaz, Seyed Mohammad Nabavi, Anupam Bishayee

**Affiliations:** 1Pharmaceutical Sciences Research Center, Health Institute, Kermanshah University of Medical Sciences, Kermanshah 6715847141, Iran; 2Department of Biochemistry, University of Allahabad, Allahabad 211 002, India; amitfbs21@gmail.com (A.K.S.); rameshbiochem91@gmail.com (R.K.); akpandey23@rediffmail.com (A.K.P.); 3Lake Erie College of Osteopathic Medicine, Bradenton, FL 34211, USA; CCroley48578@med.lecom.edu; 4Bioorganic Chemistry Laboratory, Facultad de Ciencias Básicas y Aplicadas, Campus Nueva Granada, Universidad Militar Nueva Granada, Cajicá 250247, Colombia; Ericsson.coy@unimilitar.edu.co; 5Research Institute of Biotechnology and Medical Converged Science, Dongguk University-Seoul, Ilsandong-gu, Gyeonggi-do 10326, Korea; jkpatra@dongguk.edu (J.K.P.); gitishreedas@yahoo.co.in (G.D.); 6Post Graduate Department of Biotechnology, Utkal University, Vani Vihar, Bhubaneswar 751 004, Odisha, India; routgeorgekerry3@gmail.com; 7Department of Pharmacy, University of Naples “Federico II”, Via Domenico Montesano 49, 80131 Naples, Italy; giuseppe.annunziata@unina.it (G.A.); giancarlo.tenore@unina.it (G.C.T.); 8Department of Pharmacy, Abdul Wali Khan University, Mardan 23200, Pakistan; hkdr2006@gmail.com; 9Department of Pharmacy and Biotechnology, Alma Mater Studiorum-University of Bologna, 40126 Bologna, Italy; matteo.micucci2@unibo.it (M.M.); roberta.budriesi@unibo.it (R.B.); 10Medicinal Plants Research Center, Institute of Medicinal Plants, Karaj 141554364, Iran; saeideh58_momtaz@yahoo.com; 11Pharmaceutical Sciences Research Center, The Institute of Pharmaceutical Sciences, Tehran University of Medical Sciences, Tehran 141556451, Iran; 12Applied Biotechnology Research Center, Baqiyatallah University of Medical Sciences, Tehran 1435916471, Iran; Nabavi208@gmail.com

**Keywords:** flavonoids, metabolic disorders, inflammation, oxidative stress, nuclear factor-κB, inflammatory mediators

## Abstract

A balanced metabolic profile is essential for normal human physiological activities. Disproportions in nutrition give rise to imbalances in metabolism that are associated with aberrant immune function and an elevated risk for inflammatory-associated disorders. Inflammation is a complex process, and numerous mediators affect inflammation-mediated disorders. The available clinical modalities do not effectively address the underlying diseases but rather relieve the symptoms. Therefore, novel targeted agents have the potential to normalize the metabolic system and, thus, provide meaningful therapy to the underlying disorder. In this connection, polyphenols, the well-known and extensively studied phytochemical moieties, were evaluated for their effective role in the restoration of metabolism via various mechanistic signaling pathways. The various flavonoids that we observed in this comprehensive review interfere with the metabolic events that induce inflammation. The mechanisms via which the polyphenols, in particular flavonoids, act provide a promising treatment option for inflammatory disorders. However, detailed clinical studies of such molecules are required to decide their clinical fate.

## 1. Introduction

Multicellular organisms fight infections, manage different external and internal damages, and maintain the body’s energy balance, particularly under energy deficit conditions. In overall summation, the immune system and metabolic pathways are among the primary essentialities, without which the animal kingdom would cease to exist [1]. Furthermore, both the immune and metabolic pathways co-evolved in a manner in which they are closely linked and interdependent. Nutritional imbalances disrupt metabolism, leading to irregular immune function and an elevated risk for inflammatory-associated disorders [2]. In ancient history, inflammation was characterized based on the visual observations of five cardinal signs specifically named as rubor (redness), tumor (swelling), calor (heat), dolor (pain), and lastly function laesa (loss of function) [3]. In short, inflammation could be described as a response of living tissue to local injury [4,5]. There is evidence to support that inflammation plays a decisive role in neoplastic progression. This concept is based on the relationship between incessant inflammatory activities due to viruses, bacteria, parasites, infections, and carcinogenesis and their effects within the organs and tissues [6,7,8,9,10].

Inflammation can be further divided into acute and chronic. The local effects of acute inflammation include modifications of metabolic and functional activities of polymorphonuclear cells and macrophages. The systemic effects of acute inflammation include modification of immune response and non-specific defenses against infection and neoplasia [11]. Acute inflammation can also lead to fever and leukocytosis [4]. Chronic inflammation is characterized by the medical condition of chronic inflammatory disease. This medical condition can be defined as a lengthy and persistent pro-inflammatory state marked specifically by the formation of new connective tissue [12]. There are many diseases included in this category: autoimmune disease, metabolic syndromes, neurodegenerative disease, chronic inflammatory bowel disease, chronic obstructive pulmonary disease, and cardiovascular disease (CVD) [12,13]. Among the abovementioned diseases and syndromes, metabolic syndromes are strongly associated with chronic inflammation [1]. Inflammatory reaction pathways involve various receptors and molecules, such as Toll-like receptors (TLRs) or nucleotide oligomerization domain (NOD)-like receptors (NLRs). These receptors activate major mitogen-activated protein kinase (MAPK) cascades and stimulate translocation of regulatory nuclear factor kappa-light-chain-enhancer of activated B cells (NF-κB) [7]. It was documented that the expression of pro-inflammatory cytokine mediators orchestrates an indispensable role in various CVDs, metabolic syndromes, and atherosclerosis [14,15,16,17,18].

A number of investigators reported the natural amelioration of inflammation via the use of polyphenols due to their anti-inflammatory activities [19,20,21]. It was found that numerous polyphenols, such as flavonoids, are active suppressors of inflammatory cytokines, modulators of transcription factors and inflammation-related pathways, and reducers of accumulated of nitric oxide (NO) or reactive oxygen species (ROS) [22]. For example, a species-specific flavonoid, glabridin, was found to attenuate mediators of inflammation including nitric oxide (NO), tumor necrosis factor-α (TNF-α), and interlukin (IL)-1β in THP-1 cells, RAW 264.7 cells, and J774a.1 cells [23,24,25,26]. Another investigator showed that glabridin also inhibited the maturation of dendritic cells by blocking NF-κB and MAPK signaling cascade [27]. Similarly, phloretin, another important polyphenol, was found to inhibit expression of IL-8, C–X–C motif chemokine 10, and TNF-α messenger RNAs (mRNAs) [28]. 

Oxidative stress and inflammatory mediators are well known for their role in generating ROS and reactive nitrogen species (RNS) by affecting NADPH oxidase and nitric oxide synthase (NOS) respectively. ROS trigger redox-sensitive kinases, such as apoptosis signal-regulating kinase 1 (ASK1), which in turn activates downstream MAPKs, NF-κB, and activator protein 1 (AP-1), resulting in the induction of inflammatory gene expression. Research shows that phenolic phytochemicals possess strong antioxidant activity due to the presence of hydroxyl groups within their aromatic rings [29,30]. The molecular mechanism involved lies in the capacity of the phenolic phytochemicals to increase the level of anti-inflammatory genes, such as glutathione peroxidase (GPx), superoxide dismutase (SOD), and hemeoxygenase-1 (HO-1), via activation of the nuclear factor erythroid 2-related factor 2 (Nrf2) [30]. 

To manage metabolic syndromes, global strategies initially focused on lifestyle changes such as diet and physical activity. Diets rich in plant-derived products with a high content of bioactive compounds, mono/polyunsaturated fatty acids, and polyphenols were shown to lower related risks of metabolic syndrome [31,32,33]. Polyphenols are available in various regular diets, indicating negligible side effects if taken in a controlled manner. This fact makes them a good candidate for their use in various clinical trials in treatment of various diseases. There is a large body of evidence to suggest that polyphenols play an effective role as anti-inflammatory agents in various metabolic diseases [34,35,36]. However, there is no review article that specifically concentrates on this subject. In addition, there is little information on the utilization of polyphenols to ameliorate the effect of metabolic disorders, as well as their in vitro and in vivo evaluation.

The purpose of this review is to summarize recent data on dietary polyphenols, with a special emphasis on flavonoids, which affect the inflammation involved in various metabolic disorders, and to gather the results of clinical studies on the chronic supplementation of flavonoids on metabolic syndromes features.

## 2. Inflammation and Metabolic Disorders

Inflammation is a defensive strategy developed in higher organisms in reaction to the harmful effects of tissue injury, microbial infection, and other detrimental conditions. It is an important immune response by the host that aids in tissue repair and the elimination of harmful stimuli [37]. However, long-term inflammation is often harmful and can cause several metabolic diseases [13] (Figure 1).

Moreover, high caloric intake coupled with inactive lifestyle leads to increased incidence of obesity, type 2 diabetes, and cardiovascular diseases, which are associated with chronic diseases [38,39,40]. Metabolism and inflammation are interrelated. Metabolic disorders display a strong inflammatory foundation, and inflammation is also linked to metabolic changes. The complex interaction between inflammatory and metabolic pathways is underscored by the biological and functional similarity of macrophages and adipocytes. In both cell types, similar gene expression is observed. Several adipocyte proteins are expressed by macrophages such as peroxisome proliferator-activated receptor-γ (PPARγ), adiponectin protein 2 (aP2), and adipocyte/macrophage fatty-acid binding protein (FABP). In addition, adipocytes express many “macrophage” gene products, such as IL-6, TNF-α, and matrix metalloproteinases (MMPs) [41,42]. 

Monocytes at infection sites differentiate into macrophages under the influence of the local cytokine environment [43]. Based on function and cytokine expression, differentiated macrophages are classified as M1 and M2 macrophages. M1 macrophages produce pro-inflammatory cytokines such as IL-1, IL-6, and TNF-α, which result in the mobilization of neutrophils and help to initiate the innate immune response against pathogens. Furthermore, T cells are stimulated by the cumulative action of pro-inflammatory cytokines and their ligands. M1 macrophages and T lymphocytes are linked to chronic inflammation in metabolic disorders. M2 macrophages are anti-inflammatory mediators that participate in wound healing and the angiogenic process [44]. 

The transcription factor NF-κB is a well-known inflammatory and immunological mediator that connects inflammation to metabolic responses. It helps to better understand metabolic diseases and provides insight into the development of therapeutic strategies. Pathogen-associated molecular patterns (PAMPs) and cytokines trigger cell surface receptors, including TLRs, which initiate a signaling pathway that activates NF-κB [45]. NF-κB-regulated gene expression is responsible for the differentiation of various types of immune cells. Involvement of NF-κB was revealed in three common metabolic disorders: atherosclerosis, obesity, and insulin resistance. Extracellular stimuli control the activity of NF-κB. The nuclear localization and transcriptional function of NF-κB is prevented by inhibitor of κB (IκB). IkB sequesters NF-κB in the cytoplasm of resting cells. Interaction of cytokine or PAMP with cell surface receptors begins a signaling sequence that activates the inhibitor of κB kinase (IKK) complex. Phosphorylation of IκB by IKK stimulates its break down that releases NF-κB and, consequently, NF-κB moves inside the nucleus and promotes transcription of target genes. Research findings suggest that the IKKγ/IKKβ complex initiates NF-κB-regulated gene expression downstream of TLRs and cytokine receptors [45].

Most resting cells do not express the IKK3, but NF-κB induces its transcription downstream of inflammatory stimuli [46]. Research findings showed the involvement of the non-canonical IKK kinase IKK3 in high-fat diet-induced obesity. In response to excess nutrients, IKK3 expression increases approximately 40 times in fat cells and fat-infiltrating macrophages [47]. 

Deficiency of IKK3 causes uncoupling of obesity from a fat-rich diet by increasing energy consumption, thermogenesis, and aerobic respiration. IKK3 knockdown mice are protected from pro-inflammatory pathway activation, chronic inflammation in liver and adipose tissue, and diet-induced insulin resistance because of lesser weight gain and possibly improper cytokine production and signaling pathway [48,49,50,51,52]. Hence, IKK family signaling cascade performs a vital function in overnutrition-induced obesity, as well as in metabolic disease. 

Inflammatory processes are recognized to induce atherosclerosis. Atherosclerosis begins with the accumulation of excess unmetabolized lipoproteins in the blood. Oxidized lipoproteins induce vascular endothelia to release chemokines, macrophage inflammatory protein-1α and monocyte chemoattractant protein-1 (MCP-1), which attract leukocytes to the inflammation site [48,49]. Association of oxidative stress and mitochondrial dysfunction with inflammation is implicated in type 2 diabetes, leading to insulin resistance in muscle and adipocyte cells, as well as impaired insulin secretion by the β cells of the islets of Langerhans [50,51]. Oxidative stress activates NF-κB and c-Jun N-terminal kinase (JNK) pathways and, thus, shows the ability of ROS to cause insulin resistance [52,53,54].

### 2.1. Inflammation in Cardiovascular Disease

Heart disease and stroke are responsible for many of deaths yearly. Hypertension, dyslipidemia, diabetes mellitus (DM), visceral obesity, and smoking are cardiovascular risk factors which contribute to cardiovascular disease [55]. Lipid-lowering drugs in combination with blood pressure-lowering pharmacological agents and anti-thrombotic drugs can reduce the risk of atherosclerotic vascular disease. In spite of these therapeutic advances, fresh approaches are still required to manage the increasing incidence of atherosclerosis. Immunity (innate and acquired) plays an essential role in the beginning, advancement, and instability of the atherosclerotic plaque development. However, for the prevention of CVD, it is very difficult to target a single pathway due to the considerable complexity of the immune system network [56].

Inflammation is related to the development of several cardiovascular complications. Increased levels of inflammatory markers, such as C-reactive protein (CRP) and serum amyloid A (SAA), are suggested as predictive markers of future cardiovascular disease development [57]. A drawback related to the currently used biomarkers of inflammation in CVD is their non-specificity (e.g., CRP). Therefore, a major challenge for researchers is to identify specific markers of inflammation that are relevant to the pathophysiology of disease and that can be used to guide targeted therapies [57].

#### 2.1.1. Acute Myocardial Infraction (AMI)

Coronary atherosclerotic plaque rupture leads to the speedy formation of a thrombus in the infarct-associated epicardial artery and is coupled with the inhibition of blood supply distal to the obstruction site, resulting in an AMI. Plaque erosions and calcific nodules were also suggested as other causes of AMI [58]. An organized immune response begins as a result of myocardial ischemia and occurs locally at the injury site, as well as in the systemic circulation and at distant sites [57]. The immune reactions serve dual roles as mediators of injury and, afterward, as facilitators of repair and recovery. Continuous rising levels of nonspecific inflammatory mediators, e.g., IL-6 and high-sensitivity CRP in plasma, are observed in severe cases [59,60].

##### Inflammatory Pathway in AMI

Myocardial ischemia causes injury to cardiomyocytes and, if ischemia persists for a long duration, death ensues with the liberation of intracellular contents, together with alterations in the extracellular matrix. Subsequently, endogenous signals initiate a cytokine “burst”, followed by platelet activation-mediated leukocyte stimulation and tissue permeation. This further leads to the quick mobilization of neutrophils to the site of myocardial injury followed by their degranulation, which causes direct damage to endothelial cells along with production of cytokines, ROS, and proteolytic enzymes [61]. As a result, the expression of leukocyte and platelet adhesion molecules elevates in endothelial cells, which further accelerates the binding and transmigration of monocytes in circulation [62]. This process overlaps with the reduction of in situ neutrophils because of local cell death coupled with egression of cells [63]. 

Numerous endogenous signals, namely, low-molecular hyaluronic acid, heat-shock proteins, fibronectin fragments, and high-mobility group protein B1, are frequently denoted as “danger-associated molecular patterns” (DAMPs). These DAMPS can trigger innate immune pathways, comprising the complement cascade, TLRs, ROS production, and nucleotide-binding domain leucine-rich repeat-containing receptors (NLRs). The specific TLR expression in cardiomyocytes (TLR4), an important characteristic feature of activated macrophages, is considerably elevated during interaction with DAMPs [57].

Animal experiments showed that reduction in infarct size and systemic inflammation, as well as improved left ventricular remodeling, is associated with TLR4 deficiency [64,65]. Subsequent to AMI, activation of TLR4 in patient monocytes suggests progression toward heart failure. On the other hand, TLR2 deficiency is related to decreased myocardial fibrosis after AMI and enhanced left ventricular remodeling. It appears that TLR4 plays a vital role in the initial inflammatory response, while TLR2 is crucial to left ventricular remodeling and repair [66]. Amongst various NLRs, the nucleotide oligomerization domain (NOD or NACHT), leucine-rich repeats (LRR) and pyrin domain (PYD) domains-containing protein 3 (NLRP3)–inflammasome complex is recognized as a key facilitator of injury after myocardial ischemia. In addition to NLRP3, the complex includes the cysteine protease caspase-1 and the apoptosis-associated speck-like protein containing a caspase recruitment domain (CARD). The activated NLRP3–inflammasome binds and stimulates caspase-1, which transforms IL-1β to an active state, and is a key factor in the activation of various cytokines [67]. 

The complement cascade plays a major role in steering neutrophil and monocyte accumulation deeply into the damaged myocardium via the downstream complement effectors C3a and C5a. In addition, it also helps macrophages, microvascular endothelial cells, and vascular smooth muscle cells (VSMCs) in releasing monocyte chemotactic protein 1, which is also known as C–C motif chemokine 2 (CCL2) [68]. After AMI, ischemic cardiomyocytes release ROS immediately, which induce elevated levels of leukocyte chemokine, leading to an augmented capability of endothelial intercellular adhesion molecule1 (ICAM1) ligands to bind to neutrophils, activation of complement, and the accumulation of extra subpopulations of leukocytes that consequently become activated. Many cytokines, such as TNF-α, IL-1β, and IL-6, also act as key molecules in contributing to the pathophysiology of AMI [69].

#### 2.1.2. Atherosclerosis

Atherosclerosis is an inflammation-participating disease in all of its stages, that is, starting from initial lesions until the end-stage thrombotic complications [70]. It results from complex phenomena comprising the interaction of lipoproteins, blood cells, arterial wall components, and the immune system. The process of advanced atherogenic plaque development from the earliest foam cell formation involves interplay of endothelial cells, smooth muscle cells (SMCs), lymphocytes, monocytes, and macrophages. This interaction is regulated by several inflammatory mediators associated with chronic low-grade inflammation [71]. The adhesion of leukocytes to the innermost surface of the artery wall is prevented by to the presence of endothelial cells (ECs) [72]. The role of these inflammatory molecules as proxy biomarkers and causative agents in the atherogenic development and plaque vulnerability was studied by our groups. The intimate linkage between atherosclerosis and inflammation is supported by experimental evidence. Several randomized controlled trials revealed that routine use of statins for prevention of CVD mediates their effect through anti-inflammatory action [73].

It is evidenced that smoking, hypertension, high saturated-fat diet consumption, hyperglycemia, obesity, or insulin resistance could induce the expression of a range of cytokines including TNF-α, TNF-β, IL-1α, IL-1β, IL-6, and certain messenger cytokines, such as macrophage colony stimulating factor (M-CSE), MCP-1, IL-18, and vascular cell adhesion molecule-1 (VCAM-1), by ECs. Induction of these molecules leads to the activation and recruitment of monocytes during atherosclerosis [70,72]. VCAM-1 specifically binds to T lymphocytes and monocytes in an early stage of atherosclerotic plaque formation [74]. VCAM-1 expression is induced by oxidized lipids through a pathway mediated by NF-κB and certain pro-inflammatory cytokines, such as IL-1β and TNF-α [70,75]. Furthermore, through diapedesis, T lymphocytes and monocytes penetrate the endothelial lining and enter the intima of the vessel. A basic requirement of this process is a chemoattractant gradient, which involves C–C chemokine receptors (CCR) sensing MCP-1/CCL2, a member of the C–C chemokine family [70,76].

Inflammation-induced alterations in lipoprotein function increase the risk of atherosclerosis. There are reports that inflammation and infection have an association with low-density lipoprotein. However, the LDL levels do not steadily rise and may even decline with inflammation and infection [77]. 

Inflammation and infection are responsible for disturbing the anti-atherogenic activity of serum high-density lipoprotein (HDL), in addition to reducing the serum HDL level [77]. Moreover, inflammation and infection also unfavorably affect most of the steps in the reverse cholesterol transport pathway, which plays a vital role in averting cholesterol build-up in macrophages and, hence, reduces the atherosclerotic process [78]. The initial event under the influence of inflammatory cytokines starts with the decreased production of the key protein component of HDL, Apo A-1. Pro-inflammatory cytokines in macrophages further reduce the expression of apolipoprotein E, scavenger receptor class B type I, ATP-binding cassette transporter A1 (ABCA1), and ATP Binding cassette subfamily G member 1, which in turn causes a reduction in the phospholipid and cholesterol efflux from the macrophage to HDL. The transformed HDL molecules synthesized during inflammation are poor acceptors of cellular cholesterol, and they may actually perform the opposite action by delivering cholesterol to the macrophage [79,80]. 

SR-B1 also facilitates cholesterol uptake by hepatocytes from HDL particles. Its expression in the liver is decreased by pro-inflammatory cytokines. Paraoxonase, an HDL-associated enzyme, plays a major role in the prevention of LDL oxidation. Oxidized LDL is pro-atherogenic and is more easily taken up by macrophages. Inflammatory response decreases the level of paraoxonase 1 expression in the liver and brings about a decline in circulating paraoxonase activity [77]. 

Aberrant stress activates the endothelium. It results in enhanced permeability to lipoproteins and increased expression of adhesion receptors, which enables deposition of lipid in the sub-endothelial space from apolipoprotein B (apoB)-containing lipoproteins. Activation of the endothelium also stimulates the accumulation of circulating monocytes originating from the spleen or bone marrow [81].

Monocyte adhesion is governed by the elevated expression of numerous cell adhesion molecules on the luminal surface of the endothelium, such as VCAM1, P-selectin, and ICAM1. This is followed by the expression of facilitators for the transmigration process belonging to three main chemokine families, i.e., CCR2, CCR5, and CX3C chemokine receptor 1 (CX3CR1) [82]. After recruitment, monocytes can directly affect the phenotype of in situ cells, differentiate, and locally multiply into distinct functional phenotypes, or terminally differentiate into macrophages. Activated macrophages take up lipids present in apoB-containing lipoproteins via several processes comprising pinocytosis of LDL, phagocytosis of aggregated LDL, uptake of altered apoB-lipoproteins by scavenger receptors, and cluster of differentiation 36-mediated uptake [57].

Continuous lipid build-up prompts their conversion into macrophage-derived foam cells. With the growing population of foam cells within arterial wall lesions, the level of accumulation surpasses the rate of clearance and, ultimately, the foam cells merge into a lipid-rich necrotic core. Interplay of cells from both the arms of immunity, e.g., the macrophages (innate immunity) and T and B lymphocytes, as well as dendritic cells (adaptive immunity), control the local inflammatory environment with the involvement of numerous cytokines in this process [83]. 

Based on their local milieu, T lymphocytes can be triggered to release pro-inflammatory T helper type 1 (Th1) cytokines (IL-1, IL-6, and TNF-α) or Th2 cytokines (IL-4, IL-10, and IL-13). The p38 MAPK/NF-κB pathways are mainly involved in IL-1 and TNF signaling. On the contrary, signal transducing protein gp130 activates IL-6 signaling, which in turn triggers Janus kinase 1 and signal transducer and activator of transcription 1 (STAT1) and STAT3, ultimately resulting in the stimulation of macrophages and endothelial cells to produce adhesion molecules and chemokines [84]. Propagation of disease is also attributed to the features within the plaque. New vessel formation in human plaques (neo-vascularization), initiating from the vasa vasorum, is theorized to cause intraplaque hemorrhage. This not only hastens plaque extension and inflammation, but also envisages impending plaque break [85]. The hypoxic environment inside the lipid-rich necrotic core induces hypoxia-inducible factor (HIF) 1α, which in turn activates vascular endothelial growth factor, ultimately leading to angiogenesis in advanced atheroma [86]. 

In both mouse and human macrophages, HIF1α and hypoxia were shown to cause altered lipid handling and suppression of cholesterol efflux in vitro via ABCA1. Furthermore, hypoxia and HIF1α were also reported as the driving force for higher glucose uptake, increased metabolic activity, and macrophage polarization in human atheroma during in vitro and ex vivo experiments [87]. 

Generally, endothelial cells (ECs) resist adhesion by leukocytes. However, several factors such as consumption of a high saturated-fat diet, hyperglycemia, hypertension, insulin resistance, obesity, or smoking can trigger the expression of adhesion molecules by ECs, thereby permitting the attachment of leukocytes to the arterial wall. 

### 2.2. Inflammation in Diabetes and Insulin Resistance

Diabetes mellitus (DM) is specifically characterized by a moderate or absolute lack of insulin, leading to hyperglycemia [1]. DM can be broadly categorized into type 1 and type 2 diabetes. The former is an autoimmune destruction of insulin-producing pancreatic β cells, and the latter is caused by insulin resistance coupled with failure of β cells to compensate [1]. Type 2 DM (T2DM) is a public health problem that achieved pandemic extent because of its increased incidence rate. [88]. The reoccurrence of hyperglycemia may give rise to a variety of complications, such as neuropathy, retinopathy, and nephropathy, as well as increased risk of cardiovascular diseases [89,90]. Factors such as diet, inactive lifestyle, obesity, age, and genetics are reported to be responsible for the development of DM and its complications [91].

In the case of type 1 diabetes, insulin regulates the blood glucose level by signaling the cells to capture sugar from the bloodstream, but this uptake is interrupted by the destruction of β cells [92]. This is brought about by both the environmental and genetic factors under a stress environment [93]. Stress triggers the production of auto-antigens such as insulin, tyrosine phosphatase, glutamic acid decarboxylase, and islet cell antigen 69 from β cells [94,95]. These auto-antigens are recognized by the auto-antibodies and auto-reactive T cells that provide specificity to autoimmune destruction [96,97]. The infiltration of macrophages, dendritic cells, and CD4^+^ and CD8^+^ T cells, as well as the balance of CD4^+^ Th1/Th2, is responsible for the damage of β cells. Antigen-presenting cells, such as macrophages, present the antigen peptides to the circulating T cells through MHC molecules in the pancreatic lymph nodes, which activate the T cells. NF-κB is involved in the activation of the auto-reactive T cells, monocytes, and dendritic cells and is activated by pro-inflammatory cytokines [98]. The activated CD4^+^ Th1 T cells secret IL-2 and interferon-γ, which in turn activate the CD8^+^ T cells and other macrophages, which secrete various inflammatory cytokines, such as interleukin-1β (IL-1β), TNF-α, and ROS. These inflammatory cytokines play an important role in β-cell destruction [99,100]. Among them, IL-1β was reported to be the most destructive since it inhibits the mitochondrial function of the cell by generating NO [101]. In addition to these contributory factors, inflammation-driven DM development is gaining interest in the scientific community. It was found that inflammatory routes are involved in the progression of type 1 and type 2 DM; however, further studies suggest that different causal inflammatory pathways are associated with each type of DM.

The perception of association of inflammation with T2DM provides an exciting and new insight for a better understanding of the disease. Inflammation is involved in both the pathogenesis and the consequences of the disease. In diabetes, polyphenols were found to play an important role in suppressing blood glucose levels, enhancing the antioxidant status in pancreatic cells, and increasing the activation of PPARγ [102]. The probable mechanisms via which polyphenols effect this condition represent their hypocholesterolemic property, hypolipidemic activity, and antioxidant and free-radical scavenging properties. The association of inflammation in metabolic disease can be seen as altered upregulated expression of the pro-inflammatory cytokine TNF-α in adipocytes of obese animals. Binding of TNF-α with its soluble receptor results in a decrease in insulin resistance in obese animals. These findings link the bridge between elevated expression level and plasma concentration of pro-inflammatory cytokines in insulin resistance. Insulin resistance is an intricate metabolic condition, which causes insensitivity to insulin, as well as its downstream metabolic actions under normal serum glucose concentrations to the three metabolic tissues, i.e., liver, skeletal muscle, and white adipose tissue [103].

Insulin resistance plays a key role in metabolic syndromes and was also shown to be associated with nonalcoholic fatty liver disease [104]. Insulin shows its effect by binding to the surface of insulin-responsive cells and, as a result, phosphorylation occurs to the activated insulin receptor and other several substrates, including insulin receptor family member (IRS). This consequently initiates the downstream signaling cascades [105]. Inflammatory signaling targets this signaling pathway downstream of the insulin receptor, which leads to the insulin resistance. TNF-α exposure to cells or increased free fatty acid (FFA) stimulates the inhibitory phosphorylation of the serine residue of IRS-1. This inhibitory phosphorylation reduces the level of tyrosine phosphorylation of IRS-1 in response to insulin, as well as affects its binding ability to the insulin receptor, thus leading to the inhibition of downstream signaling and insulin effect.

Binding of TNF-α to its receptor activates several signaling pathways, which result in the activation of many transcription factors, such as NF-κB and JNK. As soon as these transcription factors are activated, they cause phosphorylation of IRS-1 at the 307th position (serine), and this results in insulin-receptor mediated tyrosine phosphorylation of IRS-1. A study on the human aortic endothelium showed that TNF-α regulated reduced tyrosine phosphorylation, as well as downregulated expression of the insulin receptor itself [106]. Recently, IL-6 was reported to inhibit insulin signal pathways in liver cells. This effect is related to suppressor of cytokine signaling-3 (SOCS-3), a protein which shows an association with the insulin receptor. IL-6 causes inhibition of SOCS-3 auto-phosphorylation, IRS-1 phosphorylation at its tyrosine residue, binding affinity between p85 subunit of phosphoinositide-3-kinase (PI3K) to IRS-1, and subsequently activation of Akt pathway. These effects of IL-6 are reported in both in vitro HepG-2 cells and in vivo mice [107].

IL-6, a pro-inflammatory cytokine, is released by a number of tissues, mainly adipose tissue, and it causes insulin resistance by downregulating the expression of glucose transporter-4 (GLUT-4) and IRS-1. The downregulated expression of GLUT-4 and IRS-1 is driven by IL-6-mediated activation of the JAK–STAT signaling pathway and increased level of SOCS-3 suppressor [108,109]. Thus, suppression of the level of serum IL-6 could ameliorate the insulin resistance complications [110]. Other mechanisms via which IL-6 causes insulin resistance are blockade of the PI3K pathway, impaired glycogen synthesis by decreasing the expression of miR-200s, and upregulation of friend of GATA-binding factor 2 [111,112,113]. Kim et al. [114] suggested that resistance of human skeletal muscle to insulin is IL-6-driven and leads to STAT-3-mediated *TLR-4* gene expression. Insulin resistance causes a reduction in antilipolytic activity of insulin, which subsequently increases hepatic triglyceride synthesis [115]. Furthermore, the association of chronic inflammation to insulin resistance is also acknowledged. The responsible mechanisms via which chronic inflammation induces T2DM are not well understood and subject to further research. However, it was found that, in obese conditions, adipocytes synthesize and secrete pro-inflammatory cytokines (IL-1, IL-6, and TNF-α), and they are involved in several metabolic pathways related to insulin resistance, ROS production, lipoprotein lipase activity, and adipocyte function [116]. Therefore, both activated innate and acquired immunity play a vital role in the pathogenesis of diabetes, with convincing data that type 2 diabetes includes an inflammatory component [117].

### 2.3. Inflammation in Obesity

Obesity is at the center of the metabolic disorders, and it is associated with insulin resistance, CVD, atherosclerosis, type-2 diabetes, degenerative disorders, fatty liver disease, airway disease, and certain cancers [89]. Inflammation is commonly observed in obese and overweight patients. The inflammation induced by obesity is regarded as a low-grade chronic inflammation. [118,119]. As a result of inflammation, the adipose tissues release many inflammatory mediators. In many studies, the plasma concentrations of inflammatory biomarkers were found to be significantly higher than normal, non-obese subjects [120,121]. Macrophages and adipocytes secrete several proteins that modulate overall metabolic machinery, including fat storage [122,123]. The first molecular link between inflammation and obesity is TNF-α. It is still somewhat unclear what factors trigger the expression of TNF-α. However, previous documentations suggest that dietary fatty acids (FAs), such as long-chain, marine-derived *n*-3 FAs, elicit anti-atherogenic and ant-inflammatory effects [124]. It can be said that the level of TNF-α is nutritionally regulated and, more specifically, it can also be enhanced by hyperinsulinemia alone [125]. This inflammatory cytokine is predominantly secreted by monocytes and macrophages. The activation of the TNF receptor results in stimulation of NF-κB signaling via inhibitor of κB kinase β (Ikkβ) [126]. The NF-κB/Rel family includes NF-κB1 (p50/p105), NF-κB2 (p52/p100), p65 (RelA), RelB, and c-Rel [127]. These molecules represent a family of transcription factors that are normally found in the cytoplasm in an inactive state. They are associated with a regulatory protein from the inhibitors of κB (IkB) protein family, which includes IκBγ, IκBβ, IκBα, IκBε, and Bcl-3 in higher vertebrates [128]. In response to the multiple stimuli of inflammatory cytokines, the IκBα, which is bound to the p50–p56 heterodimer and the p50 homodimer, is phosphorylated by IKK. The IKK complex consists of at least three subunits, namely, kinases IKKα, IKKβ, and IKKγ [127,129]. IKK phosphorylates IκBα specifically at the NH-terminal serine residue, which is then consecutively ubiquitinated and selectively degraded by the 26S proteasome, thereby releasing NF-κB [130]. The free NF-κB then binds to κB enhancer elements of target genes and induces transcription of pro-inflammatory genes [127,131].

In addition, the contribution of a number of mediators was suggested. The adipokines such as leptin, adiponectin, IL-6, and TNF-α show prominent effects on adipocyte metabolism and utilization of insulin and, therefore, exhibit a strong association with obesity and related metabolic disease [132]. However, the inflammation induced by obesity is low-grade and primarily supported by low concentrations of inflammatory cytokines. It was observed that obesity-induced inflammation exhibits similar characteristics in molecular aspects to atherosclerosis, which is counted as one of the major complications of metabolic syndrome in line with lipid metabolism disorder and insulin resistance [133,134,135]. It is worth mentioning that the reported findings clearly demonstrate the involvement and significance of immune cells in obesity-induced inflammation. 

### 2.4. Inflammation in Fatty Liver Disease

Fatty liver disease, also identified as hepatic steatosis, is caused by the accumulation of various fatty acids, mainly triglycerides, in the liver [136]. FFAs are the predominant type of lipids in the liver (nearly two-thirds of lipids). The accumulation of lipids in the liver is known to result from elevated de novo hepatic lipogenesis, lipolysis from visceral adipose tissue, decreased secretion of lipoprotein triglycerides, and free fatty oxidation [137].

Lipid aggregation and a stressful environment in the mitochondria of liver cells lead to the production of TNF-α and ROS, which further contribute to inflammation [138]. Inflammation is an integral part of nearly all acute and chronic liver disorders including fatty liver disorders, such as alcoholic liver disease (ALD) and non-alcoholic fatty liver disease (NAFLD). Pro-inflammatory cytokines regulate important features of liver disorders, including acute phase response, acute liver failure, cholestasis, hypergammaglobulinemia, steatosis, and fibrosis development [139,140]. The acute phase response is critically regulated by inflammatory cytokines. One of the special characteristics of many chronic liver disorders is sterilized inflammation, especially ALD and NAFLD [141]. Inflammatory cytokines, e.g., IL-1family members (IL-1α, IL-1β, IL-1Rα, IL-18, IL-33, IL-36, IL-37, and IL-38), are associated with the regulation of atherosclerosis, insulin resistance, and adipose tissue inflammation, all of which are common features of NAFLD [142,143]. 

There are several adipose tissue-derived signals that are necessary for the prevention of inflammation and for proper functioning of the liver. Obesity and insulin resistance are both responsible for lipid accumulation and liver inflammation [144]. Expression of the two most important pro-inflammatory cytokines, TNF-α and IL-6, is enhanced by the elevation of fat and insulin resistance in the body [145]. The most affected cellular organelle in this hepatic steatosis disease is the endoplasmic reticulum, whose activation is directly linked to an insulin-resistant state [146]. Many factors, such as the abnormal concentration of ROS, imbalance of Ca^2+^, and accumulation of unfolded proteins, lead to the activation of endonuclease inositol-requiring protein 1 [147]. The stimulation of the protein kinase and the secretion of FFAs from adipose tissue induce JNKs or stress-activated protein kinases (SAPKs). JNK stimulation is mediated by diverse factors, including TNF-α, IL-1, epidermal growth factor (EGF), tumor growth factor-β (TGF-β), and ROS [148]. The activated JNK signals macrophage accumulation, normal T-cell expression, and secretion of AP-1 and IL-8, which further contribute to the inflammatory conditions [149]. The endoplasmic reticulum (ER) oxidative stress directly induces the activation of IκB kinase, while MAP kinase, like NF-κB-inducing kinase, plays an indirect role. The phosphorylation of IKKα and IKKβ stimulates the expression of NF-κB and furthers the NF-κB p65–p50 heterodimeric complex. The complex interacts with the inflammatory cascades via the transcription of pro-inflammatory molecules, such as NO, adhesion molecules, cyclooxygenases (COXs), cytokines (IL-6, IL-1β, TGF-β, and TNF-α), and chemokines [150].

#### 2.4.1. Acute Alcoholic Hepatitis (AAH)

Alcoholic liver disease (ALD) is characterized by diverse metabolic features, including neutrophilia, anorexia, fever, muscle catabolism, activation of monocytes and macrophages, and altered mineral metabolism. IL-1 and TNF-α were shown to be involved in most of these processes [142]. Acute alcoholic hepatitis (AAH) was the first disease characterized by an increase in serum IL-1 activity. Fever and neutrophilia are AAH’s distinctive clinical features [151]. Patients with severe AAH are reported to have very high (almost 10-fold) serum IL-1 activity. AAH is among the first few diseases in which elevated levels of TNF-α were detected [152]. Blood levels of TNF-α and soluble tumor necrosis factor receptors are directly correlated with endotoxemia, impaired intestinal permeability, and mortality. IL-8 expression in serum is under the direct control of IL-1 and TNF-α. Hence, serum IL-8 levels are correlated with the diagnosis of AAH patients. Since IL-8 is responsible for neutrophil recruitment into the liver, its hepatic expression correlates with the progression of patients with alcoholic hepatitis (AH) [153]. ALD patients show increased levels of pro-inflammatory cytokine IL-18 in the serum, as well as in peripheral blood mononuclear cells. Higher levels of serum IL-18 and its natural antagonist, IL-18 binding protein (IL-18BP), are related to mortality in AAH [154]. An elevated level of IL-17, another inflammatory cytokine that affects neutrophil recruitment, is also observed in human ALD. Its level is directly correlated with liver inflammation [155]. Research on ALD revealed that AAH is a cytokine-driven disorder, characterized by the detectable quantity of many pro-inflammatory mediators in the systemic circulation Therefore, targeting the cytokine pathway could be a better strategy for therapeutic intervention [139].

#### 2.4.2. NAFLD

NAFLD can be defined as a collection of fat (>5%) in hepatocytes without the intake of excessive alcohol [156,157]. More than 30% of the population of the Western world is affected by this disease, particularly obese (76%) and T2DM (50%) patients. Pathogenesis of NAFLD is complex and exerts its harmful effect on liver cells via two mechanisms, i.e., accumulation of fatty acid in liver cells and oxidative stress-mediated damage to hepatocytes [158].

Accumulation of triglycerides in liver cells occurs either due to excessive intake of saturated fatty acids and obesity or because of hyperglycemia and hyperinsulinemia as in the case of insulin resistance [159]. Hyperglycemia and hyperinsulinemia upregulate the expression of carbohydrate response element-binding protein (ChREBP) and sterol regulatory element-binding protein (SREBP-1c), respectively. This in turn activates the expression of genes associated with the FFA synthesis and also decreases β-oxidation of fatty acid [160]. Inflammatory damage to liver cells results in the increased expression of *PPAR-γ*, whose expression leads to the accumulation of FFA in the liver. Liver X receptor (LXR) upregulates the expression of *SREBP-1c* and *ChREBP*, which are genes involved in the FFA synthesis that leads to steatosis [161].

Oxidative stress is the second mechanism via which NAFLD harms the liver. During liver inflammation, hepatocytes and inflammatory cells secrete cytokines, such as IL-6, TNF-α, and ROS. [158]. All of these activities activate hepatic endothelial cells, which increase cytokine expression and finally activate hepatic stellate cells, which cause phenotypic changes related to pro-fibrinogenic and pro-inflammatory functions [162]. 

Excessive accumulation of lipids leads to hepatocyte injury and activates an inflammatory response, increasing the risk of liver disease [163]. The immune response of the liver is formed by immune cells such as dendritic cells (DCs), natural killer cells (NK), Kupffer cells, neutrophils, monocytes, and NK T cells (NKT). These immune cells commence and sustain hepatic inflammation via an assembly of chemokines and cytokines, especially IL -1β and TNF, as well as ROS [164].

The activation of hepatic inflammation, as mentioned above, is caused by the build-up of infectious and non-infectious material, which is produced during cell injury and is predicted by pattern recognition receptors (PRRs). These PRRs include NLRs, C-type lectin receptors (CLRs), TLRs, and several other receptors [165]. In the pathological condition of NAFLD, the accumulation of fatty acids causes inflammatory cascades in the hepatocytes, which results in caspase-1 activation, as well as the production of TNF-α and IL-1β. Among the different NLRs, the NLRP3 inflammasome is known as a key contributor to the pathological development of inflammatory-associated diseases. The NLRP3 inflammasome is involved in the activation of apoptosis-associated speck like protein to CARD (ASC), which can result in the release of proinflammatory cytokines, such as IL-1β and IL-18. This leads to the development of NAFLD [158,166].

## 3. Flavonoids as Anti-Inflammatory Agents in Treating Metabolic Disorders

The global flavonoids market demands were estimated to be $840.2 million in 2015 and predicted to reach above $1.06 billion in 2025 [167]. Owing to flavonoids’ availability, safety, and low cost, as well as their considerable antioxidant and anti-inflammatory activities, in addition to their wide usage in functional foods, beverages, and dietary supplements, this market is expected to grow steadily. Flavonoids represent the largest class of polyphenols, which are the most abundant plant-derived bioactive compounds. The flavonoid chemical structure presents a peculiar C6–C3–C6 backbone structure [168]. In particular, it consists of two aromatic rings (also called A and B rings) that are linked by a three-carbon-chain, generating an oxygenated heterocycle (C ring) [51]. On the basis of their heterocycle structure, flavonoids are divided into several classes, including flavanols ((−)-cathecin, (+)-gallocatechin, (−)-epicatechin, (−)-epigallocatechin, (−)-epigallocatechingallate, epigallocatechin gallate, theaflavine, and theaflavinegallate), flavones (apigenin and luteonin), flavonols (kaempherol, myricetin, quercetin, isorhemetin, and rutin), flavanones (eriodictyol, hesperidin, and naringenin), anthocyanidins (cyanidin, delphinidin, malvidin, pelargonidin, peonidin, and petunidin), and isoflavones (genistein) [51,169,170,171]. Figure 2 presents chemical structures of some of the most relevant flavonoids belonging to the aforementioned classes.

Flavonoids are ubiquitously present in plants [51,172], including several plant-based foods [51,168,170], where they mainly exist in a glucoside form [172,173] and, for this reason, they are poorly absorbed [172]. Flavonoids are widely present in plant-based diets, such as the Mediterranean diet, which is characterized by a large and daily consumption of fruits, vegetables, whole grains, extra-virgin olive oil, herbs and spices, and red wine [174]. Among plant-based foods, onion, leeks, curly kale, broccoli, blueberries, and red wine are the richest sources of flavonoids [168]. In addition, apples, grapes, citrus fruits, tea, red pepper, and cocoa are other flavonoid-rich substances [170].

The most important action of these compounds is their antioxidant activity, mainly exerted by chelating metal ions or by free-radical scavenging due to the flavonoid hydroxyl group [172], thus suggesting their pivotal role in prevention and management of several pathologies including CVD, cancer, diabetes, and chronic-degenerative diseases [51], which are induced by persistent oxidative stress [175]. Interestingly, oxidative stress, in particular the endogenous production of oxidative compounds such as ROS and RNS, is strongly related to an inflammatory status [51,175,176]. In addition, a chronic inflammatory status was recognized as a major risk factor for the pathogenesis of several metabolic disorders, including CVD and diabetes [13]. Taking into account their activity against oxidative stress, dietary antioxidants might also be considered good candidates for the management of inflammation.

The anti-inflammatory activity of polyphenols, specifically flavonoids, was evaluated by both in vitro and in vivo analysis. Although there are a number of anti-inflammatory targets and mechanisms of flavonoids, the most important is the inhibition of eicosanoid-generating enzymes, including COXs, phospholipase A2, and lipoxygenases, which decrease the concentration of prostanoids and leukotrienes [177]. Other mechanistic targets include histamine release inhibition, phosphodiesterase, protein kinases, and activation of transcriptase. Inhibition of the COX cascade is observed with the in vitro treatment of flavonoids, such as quercetin. Quercetin is a strong inhibitor of COX-2 and 5-lipoxygenase (5-LOX), which are both involved in the production of eicosanoids from arachidonic acid [178]. It was also found that citrus polymethoxy flavones inhibited the production of cytokines, TNF-α, macrophage inflammatory protein-1, and IL-10 via the activated monocytes [179]. These flavones also furthered the activation of phase II antioxidant enzyme MAP kinase, protein kinase C (PKC), and Nrf2 activity [170].

The inhibition pathway of NF-κB was widely studied using a number of flavonoids, such as fisetin, silymarin, quercetin, kaempferol, rutin, luteolin, apigenin, isoliquiritigenin, xanthohumol, and chrysin. It was found that these flavonoids inhibited NF-κB activation by inhibiting a particular step in the NF-κB activation pathway. It is known that NF-κB is found in the cytoplasm associated with the IκB protein. Upon receiving signals, certain genes such as NF-κB-inducing kinase (*NIK*), mitogen activated protein kinase kinase (*MEKK*), interleukin-1 receptor-associated kinase (*IRAK*), TNF receptor-associated factor (*TRAF*), *PKC*, and *VCAM* lead to the activation of IKK. In the presence of various flavonoids, the activation of the aforementioned genes can be inhibited or mediated to inhibit the activation of IKK. Some flavonoid, such as fisetin and apigenin, can inhibit the IKK complex formation. Silymarin, quercetin, and isoliquiritigenin can inhibit activation of NF-κB transcription. Isoliquiritigenin can also inhibit NF-κB–IκB complex formation. Morin and rutin inhibit IκB, while apigenin, silymarin, kaempferol, and isoliquiritigenin inhibit phosphorylation of IκBα. Ubiquitination of IκB via the ubiquitine ligase system is inhibited by apigenin. The degradation of IκB by 26S proteasome (26S) is inhibited by quercetin and isoliquiritigenin. Translocation of activated NF-κB into the nucleus can be inhibited by isoliquiritigenin. Finally, the interaction of NF-κB with the κB binding sequence to enhance NF-κB-regulated genes may be inhibited by apigenin [180]. Similarly, various MAP kinase or JNK pathways are regulated by flavonoids which control inflammation by inhibiting Jun/ AP-1 or activation of Nrf2 and Kruppel-like factor 2 [181,182].

In some aspects, flavonoids seem to act through mechanisms similar to those of some anti-inflammatory drugs. In addition, flavonoids are able to inhibit aggregation and adhesion of platelets [183]. Interestingly, flavonoids were also demonstrated to be effective in inhibiting poly(ADP-ribose) polymerase 1 (PARP-1) [184,185], which is strongly involved in acute and chronic inflammation by acting as an upregulator in several pro-inflammatory pathways [186]. Further in vitro evidence showed a marked effect of flavonoids in suppressing several inflammatory biomarkers’ levels [187,188]. In this context, various flavonoids are reported to exert anti-inflammatory activity, including quercetin, kaempferol, catechins, morin, myricetin, apigenin, luteolin, genistein, silybin, and hesperidin [51,170].

### 3.1. In Vitro Studies on Flavonoids as Anti-Inflammatory Agents in Treating Metabolic Disorders

Although in vitro evidence showed the anti-inflammatory potential of flavonoids [189], human clinical trials are scarce and provide contrasting evidence [170]. This is probably due to the study design used or the outcomes evaluated [173,176]. Interestingly, in 2011, a cross-sectional study was conducted on a large number of healthy women from the Nurses’ Health Study cohort in order to investigate the effects of flavonoids on inflammation and endothelial function markers [190]. In particular, flavonoid intake was assessed by using a food frequency questionnaire and divided into six subclasses (flavones, flavan-3-ols, flavonols, anthocyanidins, polymeric flavonoids, and flavanones). The first significant result obtained was an inverse correlation between several classes of flavonoids and inflammatory markers. Particularly, after multivariable adjustment, a statically significant association was found between IL-18 and flavones, anthocyanidins, and total flavonoids (−0.069, *p* = 0.033; −0.079, *p* = 0.014; and −0.075, *p* = 0.020; respectively), VCAM-1 and flavonols (−0.078, *p* = 0.012), and tumor necrosis factor receptor (TNF-R)2 and flavanones (−0.061, *p* = 0.05) [190]. Similarly, Steptoe demonstrated that chronic consumption of tea significantly decreased CRP when compared to placebo (*p* = 0.05) [191].

It is well known that the anti-inflammatory activities of natural bioactive compounds, including flavonoids, play a pivotal role in the management of CVD [192]. A recent narrative review summarized the available literature about the effects of flavonoids in CVD and highlighted their action on several molecular targets involved in CVD, including reduction in the expression of cytokines, adhesion molecules, and other inflammation markers, such as CRP [193].

Many studies investigated the flavonoids anti-inflammatory activity. In an in vitro study carried out on human aortic endothelial cells, it was shown that flavonoids, in particular flavanols (including kaempferol, quercetin, and galangin) and hydroxyl flavones (including apigenin and chrysin), were effective in inhibiting the expression of endothelial adhesion molecules induced by TNF-α, such as intercellular adhesion molecule 1 (ICAM-1) and E-selectin [194]. Additionally, in vitro evidence also reported that quercetin was effective in reducing the transcription factor activator protein (AP)-1, resulting in decreased expression of ICAM-1 [195]. Further evidence showed that proanthocyanidins, from grape seed, acted by downregulating TNF-α-induced expression of *VCAM-1* [196]. Moreover, genistein, one of the most abundant isoflavones, was reported to be able to inhibit cytokine-induced adhesion of monocytes in human endothelial cells at physiological concentrations [197,198]. This suggests a marked anti-inflammatory mechanism of isoflavones in protection against vascular disease. Similar evidence was found during a study on animal models, in which administration of soy isoflavones was reported to be effective in reducing the risk of inflammation-related CVD by acting through *TNF-α* downregulation at the endothelial level [199].

The effects of flavonoids from cocoa on inflammatory markers were investigated in an in vitro study. In particular, the cellular effects of cocoa extract (CE), epicatechin (EC), and isoquercitrin (IQ) in equivalent concentrations were compared. CE, EC, and IQ reduced TNF-α levels in a dose-dependent manner, but the CE effect was higher than that produced by EC and lower than that produced by IQ. In addition, CE and EC were found to be effective in reducing the mRNA expression levels of TNF-α, IL-1α, and IL-6 [200] (Table 1).

### 3.2. In Vivo Studies on Flavonoids as Anti-Inflammatory Agents in Treating Metabolic Disorders

Zhu et al. [228] showed the anti-inflammatory effect of anthocyanins based on in vitro and in vivo studies. In particular, the treatment with an anthocyanin mixture (50 mg per mL) significantly decreased the production of CRP induced by IL-6 and IL-1β in HepG2 cells (*p* < 0.05) and the lipopolysaccharide (LPS)-induced expression levels of VCAM-1 in porcine iliac artery endothelial cells (PIECs) (*p* < 0.05). In addition, a 24-week administration of a purified anthocyanin mixture (320 mg per day) in moderate hypercholesterolemic subjects significantly reduced CRP (*p* < 0.001), VCAM-1 (*p* = 0.005), and IL-1β (*p* = 0.019) serum levels when compared to placebo [203]. 

In order to investigate the effects of flavonoids on human health, nutritional intervention-based clinical trials were conducted using specific foods naturally rich in flavonoids or enriched with these compounds. Among these, cocoa or chocolate are good candidates due to their high flavonoid content [229,230] and because their intake can be considered as an easy intervention in clinical trials. In a randomized, cross-over, placebo-controlled trial, it was observed that four-week consumption of 15 g of cocoa product significantly reduced the levels of IL-10 and IL-1β (*p* = 0.001) in moderate hypercholesterolemic subjects [231]. According to the authors, this anti-inflammatory effect was attributed to flavonoids contained in cocoa [232]. Similarly, flavonoids from cocoa powder (40 g per day for four weeks) significantly reduced p-selectin and ICAM-1 (both *p* = 0.007) in high-CVD-risk subjects [233]. Similar results were also found by Wang-Polagruto and colleagues (2006), which showed that a six-week administration of 446 mg of total flavanols from cocoa significantly reduced VCAM-1 levels in hypercholesterolemic postmenopausal women [234]. 

An interesting pilot randomized controlled study was conducted on obese and diabetic subjects in order to investigate the effects of chocolate polyphenols on endothelial function after induced hyperglycemia [235]. In particular, 13.5 g of high-polyphenol (HP) or low-polyphenol (LP) (control) chocolate was administrated after 12 h of fasting. One hour after chocolate intake, a glucose load (75 g) was given to induce hyperglycemia, and then inflammatory markers were assessed 120 min after. Authors found that HP, when compared to control, showed reduced ICAM-1 (ng/mL, HP vs. LP; fasting: 325.6 ± 9.0, after glucose load: 310.0 ± 8.4 (*p* = 0.20) vs. fasting: 321.1 ± 7.6, after glucose load: 373.6 ± 10.5 (*p* = 0.04)), p-selectin (ng/mL, HP vs. LP; fasting: 253.0 ± 14.8, after glucose load: 235.0 ± 7.7 (*p* = 0.62) vs. fasting: 265.0 ± 15.2, after glucose load: 268.5 ± 12.4 (*p* = 0.92)), E-selectin (ng/mL, HP vs. LP; fasting: 111.3 ± 5.8, after glucose load: 96.6 ± 5.6 (*p* = 0.09) vs. fasting: 94.4 ± 4.0, after glucose load: 105.8 ± 3.5 (*p* = 0.28)), and p-selectin glycoprotein ligand 1 (U/mL, HP vs. LP; fasting: 281.9 ± 12.2, after glucose load: 212.6 ± 8.7 (*p* = 0.13) vs. fasting: 262.9 ± 5.5, after glucose load: 327.5 ± 7.3 (*p* = 0.09)) [235]. Although this pilot study was conducted on a small number of subjects, it provides evidence about the acute beneficial effects of flavonoid intake, from a dietary source, on inflammation.

In a randomized, cross-over, double-blind, placebo-controlled clinical trial, Dower and colleagues [236] investigated the effects of flavonoids on inflammation and endothelial dysfunction markers on pre-hypertensive subjects. Equimolecular amounts (345 µmol) of quercetin and epicatechin were administrated for four weeks. The serum levels of VCAM-1, ICAM-1, E-selectin, von Willebrand factor (vWf), monocyte chemotactic protein (MCP)-1 (endothelial dysfunction biomarkers), IL-1β, IL-6, IL-8, TNF-α, CRP, and serum amyloid A (SAA) (inflammation biomarkers) were evaluated. All biomarkers’ levels decreased after epicatechin and quercetin treatment, and statistically significant decreases were found in E-selectin levels by epicatechin (−7.7 ng/mL; 95% confidence interval (CI): −14.5, −0.83; *p* = 0.03) and in E-selectin and IL-1β by quercetin (−7.4 ng/mL; 95% CI: −14.3, −0.56; *p* = 0.03 and −0.23 pg/mL; 95% CI: −0.4, −0.06; *p* = 0.009, respectively) [237].

The anti-inflammatory effects of the flavonoids from red wine were highlighted in a randomized, cross-over study in which 100 mL of red or white wine was administered twice daily (20 g ethanol). After four weeks of intervention, the authors found that, in both the red and white wine groups CRP, ICAM-1, and IL-6 levels were significantly reduced (*p* < 0.01); however, VCAM-1 and E-selectin levels were significantly decreased only in the red wine group (*p* < 0.01), suggesting a greater anti-inflammatory effect of red wine due to its high polyphenol content [238]. Interestingly, a one-month administration of 100 g per day of grape seed-derived proanthocyanidins was reported to be able to reduce the levels of VCAM-1, ICAM-1, and selectin in patients with systemic sclerosis [239]. This shows the anti-inflammatory activity of this class of flavonoids.

Karlsen and colleagues [240] conducted a randomized, placebo-controlled trial to investigate the effect of bilberry juice on inflammatory biomarkers in subjects with elevated risk of CVD. They found statistically significant reduced levels of CRP, IL-6, IL-15, and monokine induced by interferon (INF)-γ in the treatment group compared to placebo (*p* = 0.027, *p* = 0.037, *p* = 0.008, and *p* = 0.047, respectively) [240]. These anti-inflammatory effects were attributed to flavonoids contained in the bilberry juice, which is particularly rich in anthocyanins. 

Strawberries are another flavonoids-rich dietary source. In particular, flavonols, flavanols, anthocyanins, tannins, and hydroxycinnamic acid derivatives are contained within strawberries [241]. The effect of a strawberry beverage on post-prandial inflammatory markers was investigated in overweight and obese (body mass index (BMI) ranged from 25 to 33.5 kg/m^2^) subjects [242]. After a seven-day run-in period, subjects consumed high-energy, high-carbohydrate, and moderate-fat test meals, accompanied with strawberry or strawberry-flavored (placebo) beverages, in random order. The blood levels of glucose, insulin, CRP, IL-6, IL-1β, plasminogen activator inhibitor (PAI)1, and TNF-α were monitored. Blood levels of anthocyanins were used as an intake marker. The authors found that, six hours after the test meal, strawberry significantly decreased the levels of CRP (3.1 ± 0.1 and 2.7 ± 0.1 mg/L, in treatment and placebo group, respectively; *p* = 0.02), IL-6 (3.1 ± 0.2 and 2.6 ± 0.2 ng/L, in treatment and placebo group, respectively; *p* = 0.05), and insulin (458.4 ± 13.9 and 402.8 ± 13.9 pmol/L, in treatment and placebo group, respectively; *p* = 0.01) [242].

It is well known that high blood glucose levels are one of the most important cardiovascular risk factors. In particular, evidence showed that periodic hyperglycemia strongly impairs the endothelial function through increasing oxidative stress [243]. Persistent oxidative stress, in turn, seems to be responsible for the impairment of β-cell function, suggesting its pivotal role in the pathogenesis of T2DM [228,244,245]. 

Hyperglycemia-induced oxidative stress also acutely increases pro-inflammatory cytokine levels [246]. In addition, insulin-resistance (IR) was reported to be strongly related to inflammation [247,248], particularly in obesity and T2DM [247]. Moreover, obesity is characterized by a chronic low-grade inflammatory status, which is recognized as one of the main causes of IR [249]. Furthermore, IR and inflammation are also involved in NAFLD, forming a vicious circle with conditions promoting this disease process [250]. In this context, the effects of flavonoids in reducing oxidative stress and/or improving IR might be considered as an interesting strategy in ameliorating inflammatory status in metabolic disorders. 

Recent systematic reviews and meta-analyses focused on the effects of flavonoids on various metabolic risk factors, including IR [236,251,252]. In particular, evidence from randomized clinical trials (RCTs) showed that isoflavones from soy are able to reduce insulin (−1.37; 95% CI, −1.92, −0.81) and IR when evaluated with the Homeostatic Model Assessment (HOMA) (−0.39; 95% CI, −0.62, −0.14) [251]. Shrime et al. (2011) showed that, in short-term studies, flavonoid-rich cocoa is effective in reducing HOMA (−0.94 points; *p* < 0.001) [252]. Similarly, Hooper et al. (2012) provided evidence that flavan-3-ols are able to reduce homeostasis model assessment of insulin resistance (HOMA-IR) (−0.67; 95% CI, −0.98, −0.36), in RCTs [236].

A double-blind, placebo-controlled RCT was carried out on postmenopausal diabetic women in order to examine the effects of flavonoids on cardiovascular risk factors, including IR. The nutritional intervention consisted of an intake of 13.5 g of flavonoid-enriched chocolate twice daily, providing 100 mg of isoflavones and 850 mg of total flavan-3-ols. Adherence was monitored by the assessment of urinary levels of flavan-3-ols and isoflavones. After 12 months, a significant reduction in IR (HOMA-IR: −0.3 ± 0.2; *p* = 0.004) was found in the intervention group when compared to placebo [253]. Further in vivo studies were conducted in order to investigate the effects of flavonoids from cocoa on IR. In particular, the authors showed that daily consumption of 100 g of dark chocolate (containing about 500 mg of polyphenols and, more specifically, 65.97 mg of epicatechin, 21.9 mg of catechin, 0.59 mg of quercetin, 0.31 mg of isorhamnetin, and 0.03 mg of kaempferol) is able to significantly reduce HOMA-IR (*p* < 0.001) and increase insulin sensitivity (*p* < 0.001) when assessed by the Quantitative Insulin Sensitivity Check Index (QUICKI) (both calculated by oral glucose tolerance test (OGTT)) and compared to 90 g of white polyphenol-free chocolate [254,255]. Desideri et al. (2012) conducted a double-blind, parallel arm study in 90 elderly subjects, evaluating the effects of flavanols on various biological markers, including HOMA-IR. Subjects were randomized in three groups: high (HF), intermediate (IF), and low flavanol consumption (LF), consisting of ~990 mg, ~520 mg, and ~45 mg per day, respectively. After an eight-week treatment, HOMA-IR was significantly reduced in a dose-dependent manner in the HF and IF groups (−1.6 ± 1.0; *p* < 0.0001 and −0.9 ± 0.2; *p* < 0.0001, respectively) but not in the LF group (−0.1 ± 0.5; *p* = 0.29), suggesting that high consumption of dietary flavonoids is effective in improving insulin sensitivity [256].

Weseler and colleagues [257] conducted an ex vivo study investigating the effects of flavonoids on inflammatory markers. In particular, blood from T2DM subjects was incubated with 10 µmol/L flavonoids (flavone, fisetin, tricetin and morin) and 30 min after cytokine release was stimulated by treatment with LPS. After 16 h of incubation, tricetin and fisetin significantly reduced IL-6 levels (− 29 ± 6% and −31 ± 5%, respectively; *p* ≤ 0.001 [257].

## 4. Clinical Trials of Flavonoids in Metabolic Disorders

Flavonoids are a widespread group possessing the phenylchromane moiety. Depending on the substitution pattern, they can be subdivided into flavonols, flavanols, flavones, flavanones, isoflavones, and anthocyanins [258]. Despite flavonoids being renowned for their antioxidant capacity/activity, they have many other capabilities. Because of this fact, flavonoids can offer health benefits possibly derived from their antioxidant properties [259]; however, several studies indicated that their nutritional importance can possibly be linked to their anti-inflammatory action. Such benefits and properties can be associated with their roles in metabolic disorders and their risk factors. However, their impact requires the examination of a multifaceted scenario; thus, basic research and clinical trials need to be conducted to assess their role in the prevention/amelioration of such diseases’ end-points [258,260,261]. Although the evidence of the therapeutic uses and/or clinical efficacy of flavonoids in metabolic disorder-associated diseases is suggestive, the lack of satisfactory clinical verification makes those results significantly limited. 

However, the current literature has some important examples of clinical trials on metabolic disorder-related diseases, such as insulin resistance, dyslipidemia, hypertension, and obesity [34,260]. Most of these studies are related to flavonoid-rich materials and supplements, which indicated good results in such disorders at different levels [176]. These products are regularly abundant in the diet and seem to be non-harmful in low to moderate doses. For instance, well-known flavonoid-rich supplements and foods were demonstrated to exhibit health benefits for cardiovascular disease risks, diabetes, and obesity in randomized controlled trials using green tea [31], black tea [262], cocoa [263,264], soy [265], cranberries [266], grape [267], olive [268], and silymarin [269], among others. Those studies evidenced that long- or short-term regular ingestion of the abovementioned supplements can improve/ameliorate/influence disease conditions when compared to controls. However, no specific flavonoid source can be considered superior to another, nor can the suitability of the use of a particular flavonoid source be considered more successful or reliable when it comes to health benefits. Clinical evidence for the effect of the isolated/purified flavonoid-like active principles is still lacking. Thus, there is a clear limitation to the advance of understanding with regard to the consistent effects, efficiency, and safety of these bioactive compounds on human metabolic disorders after consumption. 

In this context, the effect of some isolated flavonoids was also evaluated in various clinical trials. In some cases, their efficacy in such diseases was confirmed, with evidence about the important impacts on metabolic disorder-related abnormalities. However, the information/results/conclusions are conflicting even in their design, temporality, doses, and parameters assessed. There are many different types of flavonoids found in functional foods, and several clinical trial results, using different flavonoids, are presented in Table 2. In this regard, quercetin, a naturally occurring flavonol, which is found in many food products (e.g., vegetables, leaves, fruits, and grains) as the major dietary flavonoid, was evaluated in various clinical trials. Despite the promising experimental findings at in vitro and preclinical levels, randomized controlled trials resulted in mixed outcomes regarding the impact of quercetin in metabolic disorders. In fact, some trials did not suggest any clinically relevant effects of quercetin supplementation. For instance, one study showed a reduction of plasma lipid levels [270], whereas other studies indicated reasonable clinical evidence for cardioprotection by anti-inflammatory means [271]. Those studies were designed as placebo-controlled, double-blind, randomized, cross-over trials in healthy or condition-having (e.g., prehypertensive, hypertensive, obese/overweight, dyslipidemic, postmenopausal, with rheumatoid arthritis, with sarcoidosis, with metabolic syndrome light-chain (AL) amyloidosis) male/female subjects (18 to 200). The evaluating doses of quercetin ranged from 150–2000 mg/day for one day to eight weeks. Results from these trials indicated some improvement in risk factors of CVD, even as cardioprotective in CVD episodes, through recovery of endothelial function and inflammation reduction [237,271,272,273]. However, in spite of the supra-nutritional doses employed, quercetin seems to have no effects on systemic and adipose tissue inflammation, as well as insulin and glucose levels; however, it reduces elevated plasma uric acid concentrations [274,275]. Thus, no influence on innate immune function or inflammation was perceived [276]. On the other hand, it can exert slightly pro-inflammatory effects [272]. Additionally, quercetin exhibited blood pressure-lowering effects in overweight–obese carriers of the apoE3 genotype and reduced the HDL cholesterol and apoA1 levels in apoE4 subjects [277], as well as improved the symptoms and disease activity in women with rheumatoid arthritis [278]. In the case of sarcoidosis and dyslipidemia, quercetin intake was related to a decrease in oxidative stress and inflammation in sarcoidosis, as well as positive effects on blood lipids [273,279]. 

Similar trends were documented for other type of flavonoids, such as the tea flavan-3-ols epicatechin and epigallocatechin-3-gallate (EGCG). The clinical evidence (Table 2) suggests a cardioprotective effect by increasing endothelial function [237], involving specific activation of associated downstream signaling pathways and antioxidant potential as the main bioactive action [280]. Among those cross-over, double-blind, placebo-controlled, randomized studies, doses of these flavanols were found to be in the 100–1890 mg/day range, between four and 24 weeks, and in conditioned male/female patients (Table 2). Analogously, citrus glycosylated flavanones (such as hesperidin and naringin) were clinically demonstrated to diminish the plasma cholesterol in hypercholesterolemic patients, as well as improve levels of different factors in type 2 diabetes, but no effect was evidenced on blood total and LDL cholesterol (LDL-C) concentrations [281,282]. These cross-over, double-blind, placebo-controlled, randomized trials were performed using doses ranging from 400–800 mg/day in conditioned subjects.

Placebo-controlled, double-blind, randomized, cross-over studies in conditioned patients (19–138) that took place during two weeks to three years of supplementation showed that the isoflavones genistein (54 mg/day) and puerarin (500 mg/day) can improve surrogate end-points associated with risk for diabetes and cardiovascular problems. This was found even in conditions such as postmenopausal women with metabolic syndrome [283] and acute myocardial infarction or coronary heart disease [284,285]. Furthermore, genistein also enhanced insulin sensitivity indexes in patients with hyperinsulinemia [286] and showed positive effects on some cardiovascular risk factors in postmenopausal women [287]. On the other hand, clinical efficacy was not detected during intake of genistein in patients with Sanfilippo disease [288].

## 5. Concluding Remarks and Future Perspectives

The inflammatory response is initiated in response to injury and infection. According to our limited understanding on the biological intricacy of chronic inflammatory disorders, the development of anti-inflammatory therapy remains a critical challenge. It is well known that oxidative stress and inflammation elicit generation of ROS and RNS, which trigger redox-sensitive kinases, such as ASK1, which in turn activate downstream MAPKs, NF-κB, and AP-1. This results in the induction of inflammatory gene expression. Polyphenolic compounds increase the level of anti-inflammatory genes, such as *SOD*, *GPx*, and *HO-1* via activation of Nrf2 (Figure 3). Involvement of NF-κB, a well-known inflammatory and immunological mediator, was revealed in three common metabolic disorders, i.e., atherosclerosis, insulin resistance, and obesity. Flavonoids suppress inflammatory cytokines, modulate transcription factors and inflammation-related pathways, reduce the level of ROS by metal ion chelating or by radical scavenging, and inhibit eicosanoid-generating enzymes, including COXs, lipoxygenases A2, and phospholipase, thereby diminishing the concentration of prostanoids and leukotrienes. In conclusion, the current clinical synthetic drugs are not producing satisfactory results in the management of metabolic disorders induced by inflammation and are only effective in treating the disease-associated symptoms. Thus, new effective therapeutic agents that have the ability to encounter the pathology and restore the normal physiology are needed. Furthermore, experimental outcomes for several metabolic disorders also appear to vary in results and conclusions, limiting the applicability of those results to practice. However, some clinical findings indicated that quercetin and genistein seem to have cardioprotective effects and that flavanols have influences on inflammation; however, outcomes are too preliminary to be included within clinical practice. Due to this fact, further research that includes more rigorous, randomized, cross-over, placebo-controlled trials, based on a large number of patients/samples/data, and comprising both men and women, are required in order to provide clarification on affirmations regarding the healthy impact of flavonoids on metabolic disorders.

## Figures and Tables

**Figure 1 ijms-20-04957-f001:**
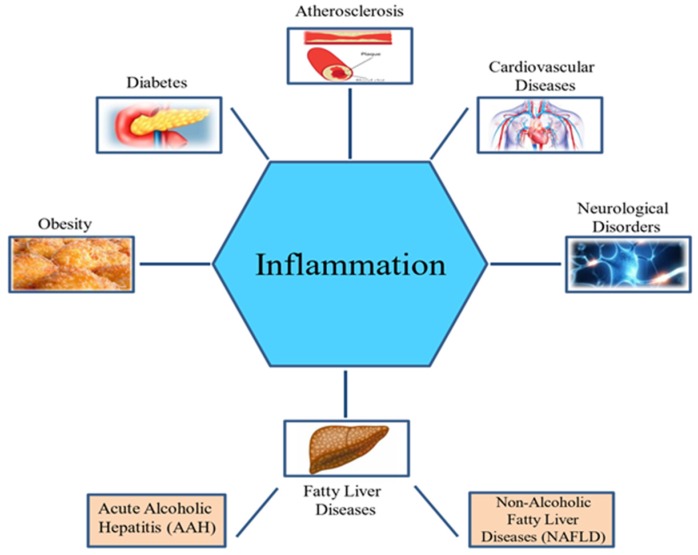
Inflammation and metabolic disorders.

**Figure 2 ijms-20-04957-f002:**
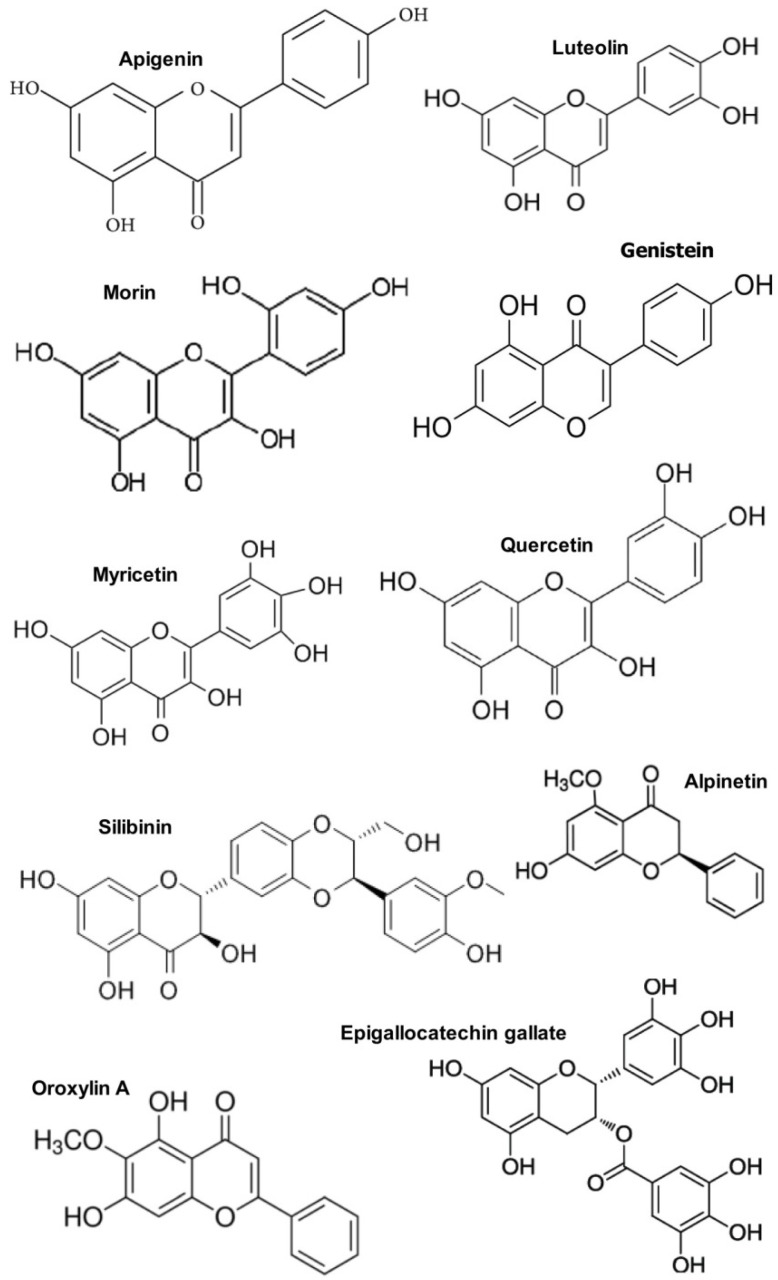
Chemical structure of most relevant flavonoids with therapeutic effects in metabolic syndrome by targeting inflammatory pathways.

**Figure 3 ijms-20-04957-f003:**
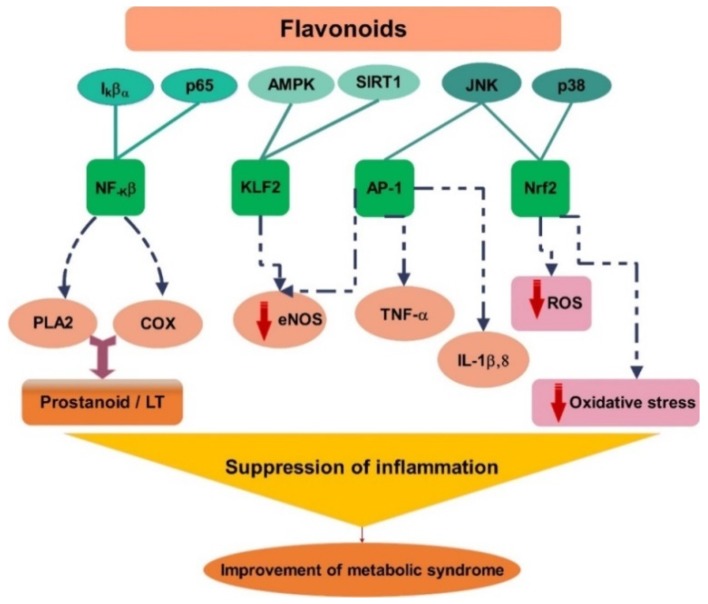
Underlying anti-inflammatory pathways involved in therapeutic effects of flavonoids in metabolic syndrome. NF-κB, nuclear factor-κB; IκB, inhibitor of κB; AMPK, activated protein kinase; SIRT1, Sirtuin 1; JNK, c-Jun N-terminal kinase; KLF2, Kruppel-like factor 2; AP-1, activator protein 1; Nrf2, nuclear factor erythroid 2-related factor 2; PLA2, phospholipase A2; COX, cyclooxygenase; eNOS, endothelial nitric oxide synthase; TNF-α; tumor necrosis factor-α; IL, interleukin, ROS, reactive oxygen species; red↓, decrease; all other ↓, direction of events.

**Table 1 ijms-20-04957-t001:** In vitro studies on polyphenols, improving metabolic disorders via ameliorating inflammatory pathways.

Polyphenols	Cell Line	Mechanism of Action	References
Flavonols (kaempferol, quercetin, and galangin)	Human aortic endothelial cells	Inhibiting the expression of intercellular adhesion molecule 1 (ICAM-1) and E-selectin	[194]
Hydroxyl flavones (apigenin and chrysin)	Human aortic endothelial cells	Inhibiting the expression of ICAM-1 and E-selectin	[194]
Quercetin	Human endothelial cells	Reducing the transcription factor activator protein (AP)-1 and the expression of ICAM-1	[195]
	Human umbilical vein endothelial cells (HUVECs)	Reducing the adhesion molecule and monocyte chemotactic protein-1 (*MCP-1*) gene expression	[201]
	Aortic endothelial cells	Inhibiting the nuclear factor kappa-light-chain-enhancer of activated B cells (NF-κB) & AP-1 DNA-binding activity	[202]
Proanthocyanidins	Human endothelial cells	Suppressing the vascular cell adhesion molecule-1 (VCAM-1) expression	[196]
Genistein	Human endothelial cells	Inhibiting the monocytes adhesion	[197,198]
Cocoa extract (CE), epicatechin (EC), and isoquercitrin (IQ)	Macrophages	1-CE, EC, and IQ reduced tumor necrosis factor- alpha (TNF-α) levels 2-CE and EC reduced the messenger RNA (mRNA) expression of TNF-α, interleukin (IL)-1α, and IL-6	[200]
Anthocyanins	HepG2 cells	Decreasing the C-reactive protein production	[203]
	Porcine iliac artery endothelial cells	Suppressing the VCAM-1 expression	[203]
Juice from bergamot (*Citrus bergamia* L.) rich in flavonoids (neohesperidin, naringin, melitidin, meoeriocitrin, and hesperetin)	THP-1 cells	Inhibiting the NF-kB activation Sirtuin 1 (SIRT1) activation	[204]
Baicalin, baicalein, or wogonin	The nucleus of HUVECs	Inhibiting vascular inflammation through decreasing the p65 NF-κB expression	[205]
Naringenin	HUVECs	Anti-inflammatory Anti-atherosclerotic Inhibiting monocyte-endothelium adhesion Decreasing chemokines levels and NF-κB nuclear translocation via reduced phosphorylation of IKKα/β, IκB-α, and NF-κB	[206]
Puerarin, nobiletin, quercetin, quercetin 7-*O*-sialic acid, luteolin, and apigenin	HUVECs	Inhibiting the NK-κB pathway	[207,208,209,210,211]
Hesperetin-3′-sulgate, hesperetin-3′-glucur-onide, and naringenin-4′-glucuronide	HUVECs	Reducing the adhesion of monocytes Inhibiting several chemokines due to NF-κB inhibition	[212]
Phytoestrogen extracts from *Glycine max* (soybean), genistein, formononetin, biochanin A, and daidzein	HUVECs	Reducing the adhesion of monocytes Inhibiting several chemokines due to NF-κB inhibition	[213]
Vitexicarpin, kaempferol, pelargonidin, epigallocatechin-3-gallate, and biochanin A	HUVECs	Inhibiting the NF-κB	[214,215,216,217,218]
A black rice (*Oryza sativa* L.) polar extract, highly rich in anthocyanin	Raw 264.7 macrophage cells	Inhibiting NF-κB and AP-1 translocation into the nucleus	[219]
*Dimocarpus longan* Lour. flowers	Raw 264.7 macrophage cells	Suppressing the NF-κB and AP-1 signaling pathways	[220]
*Cerbera manghas* L. leaves, rich in flavonoids	Raw 264.7 macrophage cells	Suppressing the c-Jun N-terminal kinase in AP-1 pathway	[221]
Kaempferol	Raw 264.7 macrophage cells	Inhibiting the AP-1 activity	[221]
Nobiletin	HepG2	Inhibiting the AP-1 activity	[222]
Berry anthocyanins	Raw 264.7 macrophage cells	Decreasing reactive oxygen species (ROS) levels through an nuclear factor erythroid 2-related factor 2 (Nrf2)-dependent mechanism	[223]
*Prunella vulgaris* L. var. lilacina, rich in flavonoids	Raw 264.7 macrophage cells	Increasing the Heme oxygenase (HO)-1 protein expression through a mechanism involving phosphoinositide 3-kinases (PI3K)/Nrf2 pathways	[224]
Isovitexin	Raw 264.7 macrophage cells	Increasing the HO-1 expression levels Increasing the activation of Nrf2	[225]
Luteolin and luteolin-7-*O*-glucoside	Raw 264.7 macrophage cells	Increasing Nrf2 activation through the regulation of p38 and c-Jun N-terminal kinase (JNK) signaling pathway	[226]
Grape seed proanthocyanidin extracts	HUVECs	Inducing the endothelial nitric oxide synthase (eNOS) expression through the increase of Kruppel-like factor 2 (KLF2) expression Increasing the 5’ AMP-activated protein kinase (AMPK) phosphorylation and SIRT1 protein level	[227]

**Table 2 ijms-20-04957-t002:** Clinical trials of pure flavonoids used in the treatment of metabolic disorder-associated diseases.

Flavonoid	Dose	Trial Type and Patients	No. of Patients	Duration of Treatment	Significant Results	Reference
Quercetin	150 mg/day	Placebo-controlled, randomized, double-blind, cross-over study in healthy volunteers with apoE genotype 3/3, 3/4 and 4/4	49	8 weeks	↓Waist circumference, postprandial systolic blood pressure (BP), and postprandial triglyceride (TG) concentrations, ↑high-density lipoprotein (HDL)-cholesterol concentrations and levels of tumor necrosis factor-α (TNF-α)	[271]
	160 mg/day	Placebo-controlled, randomized, double-blind, clinical study in healthy (pre)hypertensive men and women, aged 40–80 years, with a systolic BP = 125–160 mm Hg	26	4 weeks	No effect on flow-mediated dilation, insulin resistance, or other cardio vascular diseases (CVD) risk factors	[272]
	2000 mg/day	Placebo-controlled, randomized, double-blind, clinical study in non-smoking, untreated sarcoidosis patients, matched for age and gender	18	1 day	↑Total plasma antioxidant capacity; ↓markers of oxidative stress and inflammation in the blood of sarcoidosis patients	[273]
	160 mg/day	Placebo-controlled, randomized, double-blind, clinical study in healthy (pre)hypertensive volunteers (40–80 years)	37	4 weeks	↓Levels of sE-selectin, interleukin (IL)-1β, ↓endothelial dysfunction	[237]
	150 mg/day	Placebo-controlled, randomized, double-blind, clinical study in overweight/obese subjects aged 25–65 years with metabolic syndrome traits	93	6 weeks	↓Serum HDL-cholesterol concentrations, ↓oxidized LDL	[289]
	162 mg/day	Placebo-controlled, randomized, double-blind, clinical study in overweight-to-obese patients with pre- and stage 1 hypertension	68	6 weeks	No significant effects on parameters tested	[274]
	1095 mg/day	Placebo-controlled, randomized, double-blind, clinical study in overweight/obese men and postmenopausal women	9	1 day	Dietary fat increased the levels of methylated quercetin metabolites	[275]
	150 mg/day	Placebo-controlled, randomized, double-blind, clinical study in overweight-obese patients aged 25–65 years with metabolic syndrome traits in relation to (apo) E genotype	93	5 weeks	↓Systolic BP in the apoE3 group, ↓serum HDL cholesterol and apoA1, ↑LDL:HDL cholesterol ratio, ↓plasma oxidized LDL and tumor necrosis factor-alpha in the apoE3 and apoE4 groups	[277]
	500 mg/day	Placebo-controlled, randomized, double-blind, clinical study in healthy volunteers (19–60 years) with higher plasma uric acid concentration	22	4 weeks	↓Plasma uric acid concentrations	[290]
	-	Placebo-controlled, randomized, double-blind, clinical study in healthy patients with dyslipidemia	200	8 weeks	↓Average cholesterol, triglycerides, cholesterol, triglyceride and low-density lipoprotein (LDL) values with parallel increase in HDL	[279]
	500 mg/day;1000 mg/day	Placebo-controlled, randomized, double-blind, clinical study in healthy female subjects (aged 30–79 years)	120	12 weeks	↑Plasma quercetin at both doses	[276]
	500 mg/day	Placebo-controlled, randomized, double-blind, clinical study in women with rheumatoid arthritis (RA)	50	8 Weeks	↓Early morning stiffness (EMS, tender joint count (TJC), morning pain, and after-activity pain, ↓28-joint disease activity score (DAS-28) and health assessment questionnaire (HAQ) scores, ↓plasma hs-TNF-α level	[278]
Epicatechin	100 mg/day	Placebo-controlled, randomized, double-blind, clinical study in healthy (pre)hypertensive volunteers (40–80 years)	37	4 weeks	↓Endothelial dysfunction	[237]
	100 mg/day	Placebo-controlled, randomized, double-blind, clinical study in subjects with hypertriglyceridemia	30	4 weeks	Favorable effects on glycemia homeostasis, lipid profile and systemic inflammation	[280]
Epigallocatechin-3-gallate (EGCG)	300 mg/day	Placebo-controlled, randomized, double-blind, clinical study in obese, pre-menopausal Caucasian women	83	12 weeks	No significant effects on parameters tested	[291]
	400 m/day	Placebo-controlled, randomized, double-blind, clinical study in overweight/obese male subjects, aged 40–65 years	46	8 weeks	↓Diastolic BP	[292]
	300 mg/day	Placebo-controlled, randomized, double-blind, clinical study in overweight men	6	3 days	↓Respiratory quotient (RQ)	[293]
	1890 mg/day	Placebo-controlled, randomized, double-blind, clinical study in patients with light-chain (AL) amyloidosis	57	24 weeks	↓Urinary albumin level	[294]
	282 mg/day	Placebo-controlled, randomized, double-blind, clinical study in overweight and obese subjects	38	12 weeks	↓Visceral adipose tissue mass, ↑increased oxidative capacity in permeabilized muscle fibers, ↓fasting and postprandial respiratory quotient, ↓increase in plasma triacylglycerol concentrations	[295]
Hesperidin	600 mg/day	Placebo-controlled, randomized, double-blind, clinical study in patients with myocardial infarction	75	4 weeks	↓Serum E-selectin levels, ↑adiponectin and HDL-C concentrations, ↑interleukin (IL)-6, high sensitivity C-reactive protein (hs-CRP), leptin, and lipid profile	[296]
	800 mg/day	Placebo-controlled, randomized, double-blind, trial in moderately hypercholesterolemic men and women	136	4 weeks	No significant effects on parameters tested	[297]
	500 mg/day	Placebo-controlled, randomized, double-blind, clinical study in patients with type 2 diabetes	64	6 weeks	↑Total antioxidant capacity (TAC), ↓serum fructosamine, ↓ malondialdehyde (MDA), ↑8-hydroxy-2’ -deoxyguanosine (8-OHDG)	[281]
	500 mg/day	Placebo-controlled, randomized, double-blind, clinical study in patients with metabolic syndrome	24	3 weeks	↑Flow-mediated, ↓high-sensitivity C-reactive protein, serum amyloid A protein, soluble E-selectin	[298]
Naringin	500 mg/day	Placebo-controlled, randomized, double-blind, trial in moderately hypercholesterolemic men and women	136	4 weeks	No significant effects on parameters tested	[297]
	400 mg/day	Two group of patients: hypercholesterolemic and health	60	8 weeks	↓The plasma total cholesterol, ↓low-density lipoprotein cholesterol, ↓Apo B levels, ↑erythrocyte superoxide dismutase and catalase	[282]
Genistein	54 mg/day	Placebo-controlled, randomized, double-blind, clinical study in postmenopausal women with metabolic syndrome (MetS)	120	1 year	↓Fasting glucose, ↓fasting insulin, ↓homeostasis model assessment of insulin resistance (HOMA-IR) ↑HDL-C, ↑adiponectin, ↓total cholesterol, ↓LDL-C, ↓triglycerides, ↓visfatin, ↓homocysteine, ↓systolic and diastolic BP	[283]
	10 mg/kg/day	Placebo-controlled, randomized, double-blind,, clinical study in patients with mucopolysaccharidosis type III (MPS III) (Sanfilippo disease)	30	1 year	↓Urinary excretion of total glycosaminoglycans (GAGs), ↓plasma hydrogen sulfide (HS) concentrations	[288]
	5 mg/kg/day	Placebo-controlled, randomized, double-blind, clinical study in patients with confirmed diagnosis of MPS III (age range 2.8–19 years)	19	1 year	↓Recurrence of infections and gastrointestinal symptoms, ↑ skin texture and hair morphology	[299]
	54 mg/day	Placebo-controlled, randomized, double-blind, clinical study in normoinsulinemic and hyperinsulinemic patients.	50	24 weeks	↓Insulin basal values,↑homeostasis model index of insulin sensitivity, ↑fasting glucose levels, ↓fasting insulin, fasting C-peptide, ↑ fractional hepatic insulin extraction was shown, ↑ HDL-C levels, ↑endothelium-dependent and -independent dilatation	[286]
	54 mg/day	Placebo-controlled, randomized, double-blind, clinical study in postmenopausal women with low bone mass	138	3 years	↓Fasting glucose, ↓insulin, ↓HOMA-IR, ↓fibrinogen, ↓homocysteine	[287]
Puerarin	500 mg/day	Placebo-controlled, randomized, double-blind, clinical study in patients with acute myocardial infarction	61	2 weeks	↓Free fatty acids (FFAs), ↓matrix metallopeptidase 9 (MMP-9), ↓c-reactive protein (CRP), ↓ infarction size	[284]
	500 mg/day	Placebo-controlled, randomized, double-blind, clinical study in patients with coronary heart disease	76	3 weeks	↑Plasminogen activator inhibitor-1 (PAI-1), ↓insulin sensitivity index (ISI), ↓high density lipoprotein cholesterol (HDL-C), ↓tissue plasminogen activator (tPA), ↓FINS level, ↑ ISI ↑Total cholesterol (TC), ↑triglycerides (TG), ↑low-density lipoprotein (LDL), ↑fasting plasma glucose (FPG), ↑fasting insulin (FINS)	[285]

↓: Decrease; ↑: Increase.

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
