# Peer review of "Targeting Inflammation by Flavonoids: Novel Therapeutic Strategy for Metabolic Disorders"

_ijms, 2019, doi:10.3390/ijms20194957_

Round 1

Reviewer 1 Report

The manuscript (ID ijms-594037) is the review considering the polyphenols targeting in the inflammation, in particular in the metabolic disorders. I really believe that this strategy is important in the treatment of metabolic disorders taking into consideration that the polyphenols are so widespread in plants/food. However, the Authors focused only on the flavonoids in the manuscript whereas other classes of compounds such as phenolic acids and their conjugates (like curcumin), tannins, lignans, stilbenoids (resveratrol) which are included into polyphenols, were missed. Only curcumin was mentioned in the introduction whereas flavonoids were mainly described. Therefore, the title is not strictly adequate. The classification of compounds described in the manuscript is required to be improved. The examples of each class should have been provided consequently for each class, including flavones, flavonols, falvan-3-ols, flavanones etc. It is not known why the Authors have chosen these structures. There is no explanation which class of compounds the structures were included according to the Authors. Thus, please change the title and keywords (polyphenols, mechanistic signaling, drugs of future).

The introduction is too lengthy as well as the section “Inflammation and metabolic disorders”. There is too long way to get to the point of the manuscript. Please rearrange it, e.g. L. 102- L. 150 should be removed or moved into further sections. In the Introduction the Authors describe too much details on the biochemical mechanisms which are provided in the further sections, e. g. “Anti-inflammatory targets of flavonoids”. The introduction should contain more general information about inflammation and flavonoids as well as their role in the traditional medicine and diet. In general, there are many repetitions when the Authors describe biochemical pathways of inflammation what weaken the aim of the review such as role of polyphenols in the inflammation.

Minor comments:

53-L. 55 Please get an advise of native speaker to change the vocabulary. “NOD” I could have not found any development of abbreviation 93 Please develop the abbreviation at the beginning of sentence. 165 “defensive tactic” please change vocabulary 174 “sedentary lifestyle” please change vocabulary 182 Please rearrange the statement because it is not clear “…expressed by adipocytes namely, IL-6, …” 206 “Now freely available…” please change vocabulary 211 “IKK3” Please develop the abbreviation at the beginning of sentence. 222 “NF-κB also add to other metabolic disorders…” Please check the statement. 224-L. 225 Please check the statements “Under the influence of oxidized lipoproteins…” Please check the statement. 225 it is “MIP-1a” should be “MIP-1α” 233 “It is has been…” should be “It has been…” 243, L. 270 “amazing intricacy” “expedites” Please change vocabulary. 256 “AMI” Please develop the abbreviation 320 Please check “TNF-β” 373 “Disruptions to normal shear stress…”please check the statement 385 Please check “viz.” 389, L. 390, L. 781 it is TNF, it should be TNF-α 418 it is “βcell”, should be “β cells” 425 “sedentary routine” Please change the vocabulary. 478 it is “in-vitro”, should be “in vitro” 512 Please check spaces 525 “Ikkα, Ikkβ, IkkΥ” Please check letters size, especially “Υ” 582, L. 586 it is “TNFα”, should be “TNF-α” please revise the whole manuscript 601, L. 826, L. 769 it is more often used “Ludwig et al.” In addition, please revise the references and check the rules of references in the “Instruction for authors”. 605, L. 618 Please check spaces. 820 “HOMA-IR” please develop abbreviations.

Section “Flavonoids as anti-inflammatory agents in treating metabolic disorders”. Firstly, provide the classification of flavonoids with example compounds which are further investigated. Secondly, please rearrange the section and divide into subsections. It is difficult to find any details in this text. In fact, some studies are focused on flavonoids in general. In my opinion the strict line between in vitro and in vivo studies should be set.

Section “Anti-inflammatory targets of flavonoids” should be removed. I guess that the targets were defined in the introduction and section 2.

765 Please check it in the reference 227 “…polyphenols, in particular flavonoids [227]” I could not found it. It means rather resveratrol and anthocyanins. 905 Please add “…flavanones such as hesperidin and naringenin…” 917 Procyanidin B2 is a member of proanthocyanidins (condensed tannins) – so please provide more details when the classification is provided. 921 Please check the cell line name “RAW264.7” as in L. 933.

Table 1. and Table 2. It is advised to display them horizontally.

Table 1. Please check “Flavanols (quercetin, kaempferol, galangin)”, should be flavonols.

953 Please remove “Among plant phenolics among others [291].” 969 Please check the statement “…low-to-moderate doses.” 971 Please change the order green tea [41], black tea [295] 1016 Please check (6-83 range) (Table 1) Shouldn’t be table 2? 1017 should be “naringenin”

Table 2. Please check position 11 “500 mg/day (38) 1000 mg/day (40)”. What do 38 and 40 mean?

Author Response

The authors of this manuscript express their sincere thanks to the reviewer for the critical assessment of our work. The authors have acted upon the recommendations of the reviewers which have resulted in a significant enhancement of the quality of this manuscript. All modifications incorporated in the manuscript are highlighted using red color font. A clean copy is also provided. A “point-by-point” response to the reviewer’s comments is outlined below.

REVIEWER 1:

Major comments:

Comment 1:

The manuscript (ID ijms-594037) is the review considering the polyphenols targeting in the inflammation, in particular in the metabolic disorders. I really believe that this strategy is important in the treatment of metabolic disorders taking into consideration that the polyphenols are so widespread in plants/food. However, the Authors focused only on the flavonoids in the manuscript whereas other classes of compounds such as phenolic acids and their conjugates (like curcumin), tannins, lignans, stilbenoids (resveratrol) which are included into polyphenols, were missed. Only curcumin was mentioned in the introduction whereas flavonoids were mainly described. Therefore, the title is not strictly adequate. The classification of compounds described in the manuscript is required to be improved. The examples of each class should have been provided consequently for each class, including flavones, flavonols, falvan-3-ols, flavanones etc. It is not known why the Authors have chosen these structures. There is no explanation which class of compounds the structures were included according to the Authors. Thus, please change the title and keywords (polyphenols, mechanistic signaling, drugs of future).

Response:

We appreciate your valuable comments. We have made the following modifications:

The section related to ‘curcumin’ was omitted. The classifications were changed according to the comment. (page 13, lines 608-616). The title and keywords have been changed. We replaced ‘polyphenols’ with ‘flavonoids’.

Comment 2:

The introduction is too lengthy as well as the section “Inflammation and metabolic disorders”. There is too long way to get to the point of the manuscript. Please rearrange it, e.g. L. 102- L. 150 should be removed or moved into further sections. In the Introduction the Authors describe too much details on the biochemical mechanisms which are provided in the further sections, e. g. “Anti-inflammatory targets of flavonoids”. The introduction should contain more general information about inflammation and flavonoids as well as their role in the traditional medicine and diet. In general, there are many repetitions when the Authors describe biochemical pathways of inflammation what weaken the aim of the review such as role of polyphenols in the inflammation.

Response:

We have considered this valuable suggestion and relevant part were deleted or transferred to section 3 titled “Flavonoids as anti-inflammatory agents in treating metabolic disorders” (pages 13-14). The length of the introduction was reduced from 4 pages to 1.5 pages. The detailed discussion on various signaling pathways has been removed eliminate redundancy. In addition, as the reviewer suggested, lines 102-150 were moved to other sections of the manuscript that correspond to the disease described.

Minor comments:

Comment 1:

53-L. 55 Please get an advise of native speaker to change the vocabulary. “NOD” I could have not found any development of abbreviation 93 Please develop the abbreviation at the beginning of sentence. 165 “defensive tactic” please change vocabulary 174 “sedentary lifestyle” please change vocabulary 182 Please rearrange the statement because it is not clear “…expressed by adipocytes namely, IL-6, …” 206 “Now freely available…” please change vocabulary 211 “IKK3” Please develop the abbreviation at the beginning of sentence. 222 “NF-κB also add to other metabolic disorders…” Please check the statement. 224-L. 225 Please check the statements “Under the influence of oxidized lipoproteins…” Please check the statement. 225 it is “MIP-1a” should be “MIP-1α” 233 “It is has been…” should be “It has been…” 243, L. 270 “amazing intricacy” “expedites” Please change vocabulary. 256 “AMI” Please develop the abbreviation 320 Please check “TNF-β” 373 “Disruptions to normal shear stress…”please check the statement 385 Please check “viz.” 389, L. 390, L. 781 it is TNF, it should be TNF-α 418 it is “βcell”, should be “β cells” 425 “sedentary routine” Please change the vocabulary. 478 it is “in-vitro”, should be “in vitro” 512 Please check spaces 525 “Ikkα, Ikkβ, IkkΥ” Please check letters size, especially “Υ” 582, L. 586 it is “TNFα”, should be “TNF-α” please revise the whole manuscript 601, L. 826, L. 769 it is more often used “Ludwig et al.” In addition, please revise the references and check the rules of references in the “Instruction for authors”. 605, L. 618 Please check spaces. 820 “HOMA-IR” please develop abbreviations.

Section “Flavonoids as anti-inflammatory agents in treating metabolic disorders”. Firstly, provide the classification of flavonoids with example compounds which are further investigated. Secondly, please rearrange the section and divide into subsections. It is difficult to find any details in this text. In fact, some studies are focused on flavonoids in general. In my opinion the strict line between in vitro and in vivo studies should be set.

Response:

We wish to thank the reviewer for his/her attention to details. All these comments were considered and appropriate corrections were made throughout the text (highlighted in red-font).

53-L. 55 Please get an advise of native speaker to change the vocabulary.

Changed accordingly, page 2.

“NOD” I could have not found any development of abbreviation 93

Changed, page 2 line 78.

Please develop the abbreviation at the beginning of sentence. 165

We have included the abbveviatiuon, page 4, line 161.

“defensive tactic” please change vocabulary 174

Changed to defensive strategy developed, page 3, line 118.

“sedentary lifestyle” please change vocabulary

Changed to inactive lifestyle, page 3, line 126.

Please rearrange the statement because it is not clear “…expressed by adipocytes namely, IL-6, …

Changed to adipocytes express many “macrophage” gene products such as, IL-6, page 3, line 133.

“Now freely available…” please change vocabulary

Changed to NF-kB, consequently, NF-kB moves inside the nucleus and promotes transcription of target genes, page 4, line 155.

“IKK3” Please develop the abbreviation at the beginning of sentence

Changed to NF-kB, consequently, NF-kB moves inside the nucleus and promotes transcription of target genes., page 4, line 158.

“NF-κB also add to other metabolic disorders…” Please check the statement" Changed to NF-kB also contribute to the development other metabolic diseases, page 4, line 174. “Under the influence of oxidized lipoproteins…” Please check the statement. 225 it is “MIP-1a” should be “MIP-1α”

Changed to Oxidized lipoproteins induce vascular endothelia to release chemokines MIP-1α, page 4, line 177 to page 5, line 178.

“It is has been…” should be “It has been…”

We have made the correction, page 5, line 187.

“amazing intricacy” “expedites”

Changed to due to the considerable complexity of the immune system network, page 5, lines 197 and 198.

AMI” Please develop the abbreviation

Changed to 2.1.1. Acute myocardial infraction (AMI), page 5, lines 207-210.

Please check “TNF-β

Changed, page 6, line 274.

Disruptions to normal shear stress…”please check the statement

Changed to Aberrant stress activates endothelium, page 7, line 320.

Please check “viz.” 389, L. 390, L. 781

We have made appropriate changes, page 5, line 217; page 8, line 339.

TNF, it should be TNF-α 418

Changed accordingly throughout the text.

“βcell”, should be “β cells”

Changed throughout the text, page 5, line 181; page 17, line 791.

“sedentary routine” Please change the vocabulary

Changed to inactive lifestyle, page 9, line 378.

it is “in-vitro”, should be “in vitro” 512

Changed, page 10, line 437.

Please check spaces 525 “Ikkα, Ikkβ, IkkΥ” check letters size, especially “Υ”

Changed, page 11, lines 484 and 485.

it is “TNFα”, should be “TNF-α” please revise the whole manuscript 601, L. 826, L. 769

We have changed throughout the text.

it is more often used “Ludwig et al.

It has been replaced with “NAFLD can be defined as a….”, page 12, line 562.

820 “HOMA-IR” please develop abbreviations.

Changed to homeostasis model assessment-insulin resistance (HOMA-IR), page 18, line 807.

Section “Flavonoids as anti-inflammatory agents in treating metabolic disorders”. Firstly, provide the classification of flavonoids with example compounds which are further investigated. Secondly, please rearrange the section and divide into subsections. It is difficult to find any details in this text. In fact, some studies are focused on flavonoids in general. In my opinion the strict line between in vitro and in vivo studies should be set.

According to the main aim of this review, this section (Section 3) focused on the anti-inflammatory effect of flavonoids in general in metabolic disorders. The section has been divided into two subsections on the base of both the disease discussed and the study type (in vitro and in vivo).

Comment 2:

Section “Anti-inflammatory targets of flavonoids” should be removed. I guess that the targets were defined in the introduction and section 2.

Response:

It has been removed as per the suggestion.

Comment 3:

765 Please check it in the reference 227 “…polyphenols, in particular flavonoids [227]” I could not found it. It means rather resveratrol and anthocyanins. 905 Please add “…flavanones such as hesperidin and naringenin…” 917 Procyanidin B2 is a member of proanthocyanidins (condensed tannins) – so please provide more details when the classification is provided. 921 Please check the cell line name “RAW264.7” as in L. 933.

Response:

Please check it in the reference 227 “…polyphenols, in particular flavonoids [227]” I could not found it. It means rather resveratrol and anthocyanins.

We agree with the suggestion and accordingly revised the sentence (page 17, line 766).

Please add “…flavanones such as hesperidin and naringenin…” 917

Changed to flavanones (such as hesperidin and naringin) (page 22, lines 916 and 917.

Procyanidin B2 is a member of proanthocyanidins (condensed tannins) – so please provide more details when the classification is provided

This section has been omitted.

Please check the cell line name “RAW264.7” as in L. 933.

Changed throughout the text as Raw 264.7.

Comment 4:

Table 1.and Table 2. It is advised to display them horizontally.

Response:

While we greatly appreciate this suggestion, we believe the journal would place the table at the appropriate place and format it accordingly.

Comment 5:

Table 1. Please check “Flavanols (quercetin, kaempferol, galangin)”, should be flavonols.

Response:

We have made the correction (Table 1, page 19).

Comment 5:

953 Please remove “Among plant phenolics among others [291].” 969 Please check the statement “…low-to-moderate doses.” 971 Please change the order green tea [41], black tea [295] 1016 Please check (6-83 range) (Table 1) Shouldn’t be table 2? 1017 should be “naringenin”

Table 2. Please check position 11 “500 mg/day (38) 1000 mg/day (40)”. What do 38 and 40 mean?

Response:

953 Please remove “Among plant phenolics among others [291].”

Changed to “Favonoids are a widespread….”, page 21, line 853.

statement “…low-to-moderate doses.” 971

Changed to “…low to moderate doses….”, page 21, line 869.

Please change the order green tea [41], black tea [295] 1016

Changed to “….green tea [31], black tea [277], cocoa…..”, page 21, line 871.

Please check (6-83 range).

It has been removed.

Shouldn’t be table 2?

It has been corrected, page 21, line 886.

1017 should be “naringenin”?

Naringin is correct, page 22, line 917; please see ref. 297.

Table 2. Please check position 11 “500 mg/day (38) 1000 mg/day (40)”. What do 38 and 40 mean?

These were typographical errors which were removed.

Reviewer 2 Report

Farzei and co-workers submitted a review about the putative potential of polyphenols in inflammatory disorders. My opinion is that the review is well written and organized, but it is not  scientifically sound.

I suggest to introduce a paragraph about the economical impact of the use of this molecule in the pharmaceutical market listing the advantages and disadvantages compared to the classic drugs.

I suggest also to improve the quality of images.

Page 2 line 56-58, the period needs to be re-organized because it is very similar to line 36 of the abstract

Author Response

he authors of this manuscript express their sincere thanks to the reviewer for the critical assessment of our work. The authors have acted upon the recommendations of the reviewers which have resulted in a significant enhancement of the quality of this manuscript. All modifications incorporated in the manuscript are highlighted using red color font. A clean copy is also provided. A “point-by-point” response to the reviewer’s comments is outlined below.

REVIEWER2:

Comment 1:

Farzei and co-workers submitted a review about the putative potential of polyphenols in inflammatory disorders. My opinion is that the review is well written and organized, but it is not scientifically sound.

Response:

We wish to thank the reviewer for valuable comments. As described below, we have addressed the reviewer’s specific comments and revised the manuscript accordingly.

Comment 2:

I suggest to introduce a paragraph about the economical impact of the use of this molecule in the pharmaceutical market listing the advantages and disadvantages compared to the classic drugs.

Response:

This is an excellent suggestion. The economic impact of flavonoids has been introduced oin the revised manuscript (page 13, lines 600-603).

Comment 3:

I suggest also to improve the quality of images.

Response:

We have improved the quality of images. A new figure (Figure 1) has been added.

Comment 4:

Page 2 line 56-58, the period needs to be re-organized because it is very similar to line 36 of the abstract

Response:

In the introduction the sentence was changed to “Nutritional imbalances disrupt metabolism, leading to irregular immune function and an elevated risk for inflammatory-associated disorders [2].” (page 2, lines 58 and 59).

Round 2

Reviewer 1 Report

The Authors have not shortened the manuscript to a large extent. 116-589 The Authors describe too much details on the biochemical mechanisms. These sections are also too lengthy. Please change other keywords, e.g. drug strategy, mechanistic signaling

477-478 Please check „Ikk”, should be „IKK” 604 Please do not include procyanidin B into anthocyanidins. Procyanidin B is a B type proanthocyanidin. Its structure is (−)-epicatechin-(4β→8)-(−)-epicatechin. 604 Please check „peonidinand”

Please provide structures of compounds (Figure 2) more related to these which were mentioned in the text.

Author Response

The authors of this manuscript express their sincere thanks to the reviewer for the critical assessment of our work. The authors have acted upon the recommendations of the reviewer which have resulted in a significant enhancement of the quality of this manuscript. All modifications incorporated in the manuscript are highlighted using blue color font. A clean copy is also provided. A “point-by-point” response to the reviewers’ comments is outlined below.

REVIEWER 1:

Comment 1:

The Authors have not shortened the manuscript to a large extent. 116-589 The Authors describe too much details on the biochemical mechanisms. These sections are also too lengthy.

Response:

We greatly appreciate the reviewer’s comments. According to the reviewer’s suggestion, we have tried our best to shorten our manuscript. We sincerely believe that the current text would be appropriate, and readers would find it useful and interesting.

Comment 2:

Please change other keywords, e.g. drug strategy, mechanistic signaling

Response:

We have replaced these keywords with more appropriate and relevant ones (page 2, lines 49 and 50).

Comment 3:

477-478 Please check „Ikk”, should be „IKK”

Response:

We have made the necessary corrections (page 10, lines 428 and 429).

Comment 4:

604 Please do not include procyanidin B into anthocyanidins. Procyanidin B is a B type proanthocyanidin. Its structure is (−)-epicatechin-(4β→8)-(−)-epicatechin.

Response:

This is an excellent suggestion. We have deleted the structure as suggested (page 12, line 547). We have also modified Figure 2 (page 2) accordingly.

Comment 5:

604 Please check „peonidinand”

Response:

The word “peonidinand” has been changed to “peonidin” (page 13, line 547).

Comment 6:

Please provide structures of compounds (Figure 2) more related to these which were mentioned in the text.

Response:

In line with the excellent suggestion, we have revised the figure (page 12).